# Denoising Diffusion Implicit Models

**Jiaming Song, Chenlin Meng & Stefano Ermon**
Stanford University
{tsong,chenlin,ermon}@cs.stanford.edu

## Abstract

Denoising diffusion probabilistic models (DDPMs) have achieved high quality image generation without adversarial training, yet they require simulating a Markov chain for many steps in order to produce a sample. To accelerate sampling, we present denoising diffusion implicit models (DDIMs), a more efficient class of iterative implicit probabilistic models with the same training procedure as DDPMs. In DDPMs, the generative process is defined as the reverse of a particular Markovian diffusion process. We generalize DDPMs via a class of non-Markovian diffusion processes that lead to the same training objective. These non-Markovian processes can correspond to generative processes that are deterministic, giving rise to implicit models that produce high quality samples much faster. We empirically demonstrate that DDIMs can produce high quality samples $10\times$ to $50\times$ faster in terms of wall-clock time compared to DDPMs, allow us to trade off computation for sample quality, perform semantically meaningful image interpolation directly in the latent space, and reconstruct observations with very low error.

## 1 Introduction

Deep generative models have demonstrated the ability to produce high quality samples in many domains (Karras et al., 2020; van den Oord et al., 2016a). In terms of image generation, generative adversarial networks (GANs, Goodfellow et al. (2014)) currently exhibits higher sample quality than likelihood-based methods such as variational autoencoders (Kingma & Welling, 2013), autoregressive models (van den Oord et al., 2016b) and normalizing flows (Rezende & Mohamed, 2015; Dinh et al., 2016). However, GANs require very specific choices in optimization and architectures in order to stabilize training (Arjovsky et al., 2017; Gulrajani et al., 2017; Karras et al., 2018; Brock et al., 2018), and could fail to cover modes of the data distribution (Zhao et al., 2018).

Recent works on iterative generative models (Bengio et al., 2014), such as denoising diffusion probabilistic models (DDPM, Ho et al. (2020)) and noise conditional score networks (NCSN, Song & Ermon (2019)) have demonstrated the ability to produce samples comparable to that of GANs, without having to perform adversarial training. To achieve this, many denoising autoencoding models are trained to denoise samples corrupted by various levels of Gaussian noise. Samples are then produced by a Markov chain which, starting from white noise, progressively denoises it into an image. This generative Markov Chain process is either based on Langevin dynamics (Song & Ermon, 2019) or obtained by reversing a forward *diffusion process* that progressively turns an image into noise (Sohl-Dickstein et al., 2015).

A critical drawback of these models is that they require many iterations to produce a high quality sample. For DDPMs, this is because that the generative process (from noise to data) approximates the reverse of the forward *diffusion process* (from data to noise), which could have thousands of steps; iterating over all the steps is required to produce a single sample, which is much slower compared to GANs, which only needs one pass through a network. For example, it takes around 20 hours to sample 50k images of size $32 \times 32$ from a DDPM, but less than a minute to do so from a GAN on a Nvidia 2080 Ti GPU. This becomes more problematic for larger images as sampling 50k images of size $256 \times 256$ could take nearly 1000 hours on the same GPU.

To close this efficiency gap between DDPMs and GANs, we present denoising diffusion implicit models (DDIMs). DDIMs are implicit probabilistic models (Mohamed & Lakshminarayanan, 2016) and are closely related to DDPMs, in the sense that they are trained with the same objective function.

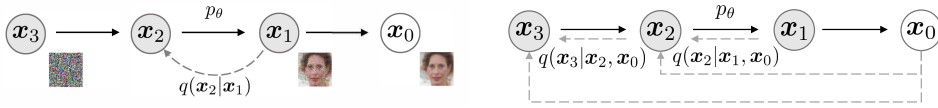

Figure 1: Graphical models for diffusion (left) and non-Markovian (right) inference models.

In Section 3, we generalize the forward *diffusion process* used by DDPMs, which is Markovian, to *non-Markovian* ones, for which we are still able to design suitable reverse generative Markov chains. We show that the resulting variational training objectives have a shared surrogate objective, which is *exactly* the objective used to train DDPM. Therefore, we can freely choose from a large family of generative models using the same neural network simply by choosing a different, *non-Markovian* diffusion process (Section 4.1) and the corresponding reverse generative Markov Chain. In particular, we are able to use *non-Markovian* diffusion processes which lead to "short" generative Markov chains (Section 4.2) that can be simulated in a small number of steps. This can massively increase sample efficiency only at a minor cost in sample quality.

In Section 5, we demonstrate several empirical benefits of DDIMs over DDPMs. *First*, DDIMs have superior sample generation quality compared to DDPMs, when we accelerate sampling by $10\times$ to $100\times$ using our proposed method. *Second*, DDIM samples have the following "consistency" property, which does not hold for DDPMs: if we start with the same initial latent variable and generate several samples with Markov chains of various lengths, these samples would have similar high-level features. *Third*, because of "consistency" in DDIMs, we can perform semantically meaningful image interpolation by manipulating the initial latent variable in DDIMs, unlike DDPMs which interpolates near the image space due to the stochastic generative process.

## 2 BACKGROUND

Given samples from a data distribution $q(\boldsymbol{x}_0)$, we are interested in learning a model distribution $p_\theta(\boldsymbol{x}_0)$ that approximates $q(\boldsymbol{x}_0)$ and is easy to sample from. Denoising diffusion probabilistic models (DDPMs, Sohl-Dickstein et al. (2015); Ho et al. (2020)) are latent variable models of the form

$$p_\theta(\boldsymbol{x}_0) = \int p_\theta(\boldsymbol{x}_{0:T}) \mathrm{d}\boldsymbol{x}_{1:T}, \quad \text{where} \quad p_\theta(\boldsymbol{x}_{0:T}) := p_\theta(\boldsymbol{x}_T) \prod_{t=1}^{T} p_\theta^{(t)}(\boldsymbol{x}_{t-1}|\boldsymbol{x}_t) \tag{1}$$

where $\boldsymbol{x}_1, \ldots, \boldsymbol{x}_T$ are latent variables in the same sample space as $\boldsymbol{x}_0$ (denoted as $\mathcal{X}$). The parameters $\theta$ are learned to fit the data distribution $q(\boldsymbol{x}_0)$ by maximizing a variational lower bound:

$$\max_\theta \mathbb{E}_{q(\boldsymbol{x}_0)}[\log p_\theta(\boldsymbol{x}_0)] \le \max_\theta \mathbb{E}_{q(\boldsymbol{x}_0, \boldsymbol{x}_1, \ldots, \boldsymbol{x}_T)} \left[ \log p_\theta(\boldsymbol{x}_{0:T}) - \log q(\boldsymbol{x}_{1:T}|\boldsymbol{x}_0) \right] \tag{2}$$

where $q(\boldsymbol{x}_{1:T}|\boldsymbol{x}_0)$ is some inference distribution over the latent variables. Unlike typical latent variable models (such as the variational autoencoder (Rezende et al., 2014)), DDPMs are learned with a fixed (rather than trainable) inference procedure $q(\boldsymbol{x}_{1:T}|\boldsymbol{x}_0)$, and latent variables are relatively high dimensional. For example, Ho et al. (2020) considered the following Markov chain with Gaussian transitions parameterized by a decreasing sequence $\alpha_{1:T} \in (0, 1]^T$:

$$q(\boldsymbol{x}_{1:T}|\boldsymbol{x}_0) := \prod_{t=1}^{T} q(\boldsymbol{x}_t|\boldsymbol{x}_{t-1}), \text{where} \quad q(\boldsymbol{x}_t|\boldsymbol{x}_{t-1}) := \mathcal{N}\left( \sqrt{\frac{\alpha_t}{\alpha_{t-1}}} \boldsymbol{x}_{t-1}, \left(1 - \frac{\alpha_t}{\alpha_{t-1}}\right) \boldsymbol{I} \right) \tag{3}$$

where the covariance matrix is ensured to have positive terms on its diagonal. This is called the *forward process* due to the autoregressive nature of the sampling procedure (from $\boldsymbol{x}_0$ to $\boldsymbol{x}_T$). We call the latent variable model $p_\theta(\boldsymbol{x}_{0:T})$, which is a Markov chain that samples from $\boldsymbol{x}_T$ to $\boldsymbol{x}_0$, the *generative process*, since it approximates the intractable *reverse process* $q(\boldsymbol{x}_{t-1}|\boldsymbol{x}_t)$. Intuitively, the forward process progressively adds noise to the observation $\boldsymbol{x}_0$, whereas the generative process progressively denoises a noisy observation (Figure 1, left).

A special property of the forward process is that

$$q(\boldsymbol{x}_t|\boldsymbol{x}_0) := \int q(\boldsymbol{x}_{1:t}|\boldsymbol{x}_0) \mathrm{d}\boldsymbol{x}_{1:(t-1)} = \mathcal{N}(\boldsymbol{x}_t; \sqrt{\alpha_t}\boldsymbol{x}_0, (1 - \alpha_t)\boldsymbol{I});$$

so we can express $\boldsymbol{x}_t$ as a linear combination of $\boldsymbol{x}_0$ and a noise variable $\epsilon$:

$$\boldsymbol{x}_t = \sqrt{\alpha_t}\boldsymbol{x}_0 + \sqrt{1-\alpha_t}\epsilon, \quad \text{where} \quad \epsilon \sim \mathcal{N}(\mathbf{0}, \boldsymbol{I}). \tag{4}$$

When we set $\alpha_T$ sufficiently close to 0, $q(\boldsymbol{x}_T|\boldsymbol{x}_0)$ converges to a standard Gaussian for all $\boldsymbol{x}_0$, so it is natural to set $p_\theta(\boldsymbol{x}_T) := \mathcal{N}(\mathbf{0}, \boldsymbol{I})$. If all the conditionals are modeled as Gaussians with trainable mean functions and fixed variances, the objective in Eq. (2) can be simplified to[1]:

$$L_\gamma(\epsilon_\theta) := \sum_{t=1}^{T} \gamma_t \mathbb{E}_{\boldsymbol{x}_0 \sim q(\boldsymbol{x}_0), \epsilon_t \sim \mathcal{N}(\mathbf{0}, \boldsymbol{I})} \left[ \left\| \epsilon_\theta^{(t)}(\sqrt{\alpha_t}\boldsymbol{x}_0 + \sqrt{1-\alpha_t}\epsilon_t) - \epsilon_t \right\|_2^2 \right] \tag{5}$$

where $\epsilon_\theta := \{\epsilon_\theta^{(t)}\}_{t=1}^{T}$ is a set of $T$ functions, each $\epsilon_\theta^{(t)} : \mathcal{X} \to \mathcal{X}$ (indexed by $t$) is a function with trainable parameters $\theta^{(t)}$, and $\gamma := [\gamma_1, \ldots, \gamma_T]$ is a vector of positive coefficients in the objective that depends on $\alpha_{1:T}$. In Ho et al. (2020), the objective with $\gamma = \mathbf{1}$ is optimized instead to maximize generation performance of the trained model; this is also the same objective used in noise conditional score networks (Song & Ermon, 2019) based on score matching (Hyvärinen, 2005; Vincent, 2011). From a trained model, $\boldsymbol{x}_0$ is sampled by first sampling $\boldsymbol{x}_T$ from the prior $p_\theta(\boldsymbol{x}_T)$, and then sampling $\boldsymbol{x}_{t-1}$ from the generative processes iteratively.

The length $T$ of the forward process is an important hyperparameter in DDPMs. From a variational perspective, a large $T$ allows the reverse process to be close to a Gaussian (Sohl-Dickstein et al., 2015), so that the generative process modeled with Gaussian conditional distributions becomes a good approximation; this motivates the choice of large $T$ values, such as $T = 1000$ in Ho et al. (2020). However, as all $T$ iterations have to be performed sequentially, instead of in parallel, to obtain a sample $\boldsymbol{x}_0$, sampling from DDPMs is much slower than sampling from other deep generative models, which makes them impractical for tasks where compute is limited and latency is critical.

## 3 VARIATIONAL INFERENCE FOR NON-MARKOVIAN FORWARD PROCESSES

Because the generative model approximates the reverse of the inference process, we need to rethink the inference process in order to reduce the number of iterations required by the generative model. Our key observation is that the DDPM objective in the form of $L_\gamma$ only depends on the marginals[2] $q(\boldsymbol{x}_t|\boldsymbol{x}_0)$, but not directly on the joint $q(\boldsymbol{x}_{1:T}|\boldsymbol{x}_0)$. Since there are many inference distributions (joints) with the same marginals, we explore alternative inference processes that are non-Markovian, which leads to new generative processes (Figure 1, right). These non-Markovian inference process lead to the same surrogate objective function as DDPM, as we will show below. In Appendix A, we show that the non-Markovian perspective also applies beyond the Gaussian case.

### 3.1 NON-MARKOVIAN FORWARD PROCESSES

Let us consider a family $\mathcal{Q}$ of inference distributions, indexed by a real vector $\sigma \in \mathbb{R}_{\geq 0}^T$:

$$q_\sigma(\boldsymbol{x}_{1:T}|\boldsymbol{x}_0) := q_\sigma(\boldsymbol{x}_T|\boldsymbol{x}_0) \prod_{t=2}^{T} q_\sigma(\boldsymbol{x}_{t-1}|\boldsymbol{x}_t, \boldsymbol{x}_0) \tag{6}$$

where $q_\sigma(\boldsymbol{x}_T|\boldsymbol{x}_0) = \mathcal{N}(\sqrt{\alpha_T}\boldsymbol{x}_0, (1-\alpha_T)\boldsymbol{I})$ and for all $t > 1$,

$$q_\sigma(\boldsymbol{x}_{t-1}|\boldsymbol{x}_t, \boldsymbol{x}_0) = \mathcal{N}\left(\sqrt{\alpha_{t-1}}\boldsymbol{x}_0 + \sqrt{1-\alpha_{t-1}-\sigma_t^2} \cdot \frac{\boldsymbol{x}_t - \sqrt{\alpha_t}\boldsymbol{x}_0}{\sqrt{1-\alpha_t}}, \sigma_t^2\boldsymbol{I}\right). \tag{7}$$

The mean function is chosen to order to ensure that $q_\sigma(\boldsymbol{x}_t|\boldsymbol{x}_0) = \mathcal{N}(\sqrt{\alpha_t}\boldsymbol{x}_0, (1-\alpha_t)\boldsymbol{I})$ for all $t$ (see Lemma 1 of Appendix B), so that it defines a joint inference distribution that matches the "marginals" as desired. The forward process[3] can be derived from Bayes' rule:

$$q_\sigma(\boldsymbol{x}_t|\boldsymbol{x}_{t-1}, \boldsymbol{x}_0) = \frac{q_\sigma(\boldsymbol{x}_{t-1}|\boldsymbol{x}_t, \boldsymbol{x}_0)q_\sigma(\boldsymbol{x}_t|\boldsymbol{x}_0)}{q_\sigma(\boldsymbol{x}_{t-1}|\boldsymbol{x}_0)}, \tag{8}$$

---

[1]Please refer to Appendix C.2 for details.

[2]We slightly abuse this term (as well as joints) when only conditioned on $\boldsymbol{x}_0$.

[3]We overload the term "forward process" for cases where the inference model is not a diffusion.

which is also Gaussian (although we do not use this fact for the remainder of this paper). Unlike the diffusion process in Eq. (3), the forward process here is no longer Markovian, since each $\boldsymbol{x}_t$ could depend on both $\boldsymbol{x}_{t-1}$ and $\boldsymbol{x}_0$. The magnitude of $\sigma$ controls the how stochastic the forward process is; when $\sigma \to \boldsymbol{0}$, we reach an extreme case where as long as we observe $\boldsymbol{x}_0$ and $\boldsymbol{x}_t$ for some $t$, then $\boldsymbol{x}_{t-1}$ become known and fixed.

## 3.2 GENERATIVE PROCESS AND UNIFIED VARIATIONAL INFERENCE OBJECTIVE

Next, we define a trainable generative process $p_\theta(\boldsymbol{x}_{0:T})$ where each $p_\theta^{(t)}(\boldsymbol{x}_{t-1}|\boldsymbol{x}_t)$ leverages knowledge of $q_\sigma(\boldsymbol{x}_{t-1}|\boldsymbol{x}_t, \boldsymbol{x}_0)$. Intuitively, given a noisy observation $\boldsymbol{x}_t$, we first make a prediction[4] of the corresponding $\boldsymbol{x}_0$, and then use it to obtain a sample $\boldsymbol{x}_{t-1}$ through the reverse conditional distribution $q_\sigma(\boldsymbol{x}_{t-1}|\boldsymbol{x}_t, \boldsymbol{x}_0)$, which we have defined.

For some $\boldsymbol{x}_0 \sim q(\boldsymbol{x}_0)$ and $\epsilon_t \sim \mathcal{N}(\boldsymbol{0}, \boldsymbol{I})$, $\boldsymbol{x}_t$ can be obtained using Eq. (4). The model $\epsilon_\theta^{(t)}(\boldsymbol{x}_t)$ then attempts to predict $\epsilon_t$ from $\boldsymbol{x}_t$, without knowledge of $\boldsymbol{x}_0$. By rewriting Eq. (4), one can then predict the *denoised observation*, which is a prediction of $\boldsymbol{x}_0$ given $\boldsymbol{x}_t$:

$$f_\theta^{(t)}(\boldsymbol{x}_t) := (\boldsymbol{x}_t - \sqrt{1 - \alpha_t} \cdot \epsilon_\theta^{(t)}(\boldsymbol{x}_t))/\sqrt{\alpha_t}. \tag{9}$$

We can then define the generative process with a fixed prior $p_\theta(\boldsymbol{x}_T) = \mathcal{N}(\boldsymbol{0}, \boldsymbol{I})$ and

$$p_\theta^{(t)}(\boldsymbol{x}_{t-1}|\boldsymbol{x}_t) = \begin{cases} \mathcal{N}(f_\theta^{(1)}(\boldsymbol{x}_1), \sigma_1^2 \boldsymbol{I}) & \text{if } t = 1 \\ q_\sigma(\boldsymbol{x}_{t-1}|\boldsymbol{x}_t, f_\theta^{(t)}(\boldsymbol{x}_t)) & \text{otherwise,} \end{cases} \tag{10}$$

where $q_\sigma(\boldsymbol{x}_{t-1}|\boldsymbol{x}_t, f_\theta^{(t)}(\boldsymbol{x}_t))$ is defined as in Eq. (7) with $\boldsymbol{x}_0$ replaced by $f_\theta^{(t)}(\boldsymbol{x}_t)$. We add some Gaussian noise (with covariance $\sigma_1^2 \boldsymbol{I}$) for the case of $t = 1$ to ensure that the generative process is supported everywhere.

We optimize $\theta$ via the following variational inference objective (which is a functional over $\epsilon_\theta$):

$$J_\sigma(\epsilon_\theta) := \mathbb{E}_{\boldsymbol{x}_{0:T} \sim q_\sigma(\boldsymbol{x}_{0:T})}[\log q_\sigma(\boldsymbol{x}_{1:T}|\boldsymbol{x}_0) - \log p_\theta(\boldsymbol{x}_{0:T})] \tag{11}$$

$$= \mathbb{E}_{\boldsymbol{x}_{0:T} \sim q_\sigma(\boldsymbol{x}_{0:T})} \left[ q_\sigma(\boldsymbol{x}_T|\boldsymbol{x}_0) + \sum_{t=2}^{T} \log q_\sigma(\boldsymbol{x}_{t-1}|\boldsymbol{x}_t, \boldsymbol{x}_0) - \sum_{t=1}^{T} \log p_\theta^{(t)}(\boldsymbol{x}_{t-1}|\boldsymbol{x}_t) - \log p_\theta(\boldsymbol{x}_T) \right]$$

where we factorize $q_\sigma(\boldsymbol{x}_{1:T}|\boldsymbol{x}_0)$ according to Eq. (6) and $p_\theta(\boldsymbol{x}_{0:T})$ according to Eq. (1).

From the definition of $J_\sigma$, it would appear that a different model has to be trained for every choice of $\sigma$, since it corresponds to a different variational objective (and a different generative process). However, $J_\sigma$ is equivalent to $L_\gamma$ for certain weights $\gamma$, as we show below.

**Theorem 1.** *For all $\sigma > \boldsymbol{0}$, there exists $\gamma \in \mathbb{R}_{>0}^T$ and $C \in \mathbb{R}$, such that $J_\sigma = L_\gamma + C$.*

The variational objective $L_\gamma$ is special in the sense that if parameters $\theta$ of the models $\epsilon_\theta^{(t)}$ are not shared across different $t$, then the optimal solution for $\epsilon_\theta$ will not depend on the weights $\gamma$ (as global optimum is achieved by separately maximizing each term in the sum). This property of $L_\gamma$ has two implications. On the one hand, this justified the use of $L_1$ as a surrogate objective function for the variational lower bound in DDPMs; on the other hand, since $J_\sigma$ is equivalent to some $L_\gamma$ from Theorem 1, the optimal solution of $J_\sigma$ is also the same as that of $L_1$. Therefore, if parameters are not shared across $t$ in the model $\epsilon_\theta$, then the $L_1$ objective used by Ho et al. (2020) can be used as a surrogate objective for the variational objective $J_\sigma$ as well.

## 4 SAMPLING FROM GENERALIZED GENERATIVE PROCESSES

With $L_1$ as the objective, we are not only learning a generative process for the Markovian inference process considered in Sohl-Dickstein et al. (2015) and Ho et al. (2020), but also generative processes for many non-Markovian forward processes parametrized by $\sigma$ that we have described. Therefore, we can essentially use pretrained DDPM models as the solutions to the new objectives, and focus on finding a generative process that is better at producing samples subject to our needs by changing $\sigma$.

---

[4]Learning a distribution over the predictions is also possible, but empirically we found little benefits of it.

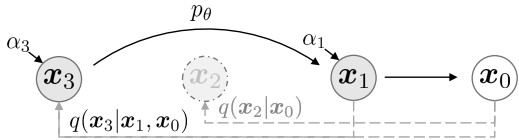

Figure 2: Graphical model for accelerated generation, where $\tau = [1, 3]$.

## 4.1 DENOISING DIFFUSION IMPLICIT MODELS

From $p_\theta(\boldsymbol{x}_{1:T})$ in Eq. (10), one can generate a sample $\boldsymbol{x}_{t-1}$ from a sample $\boldsymbol{x}_t$ via:

$$\boldsymbol{x}_{t-1} = \sqrt{\alpha_{t-1}} \underbrace{\left( \frac{\boldsymbol{x}_t - \sqrt{1 - \alpha_t} \epsilon_\theta^{(t)}(\boldsymbol{x}_t)}{\sqrt{\alpha_t}} \right)}_{\text{`` predicted } \boldsymbol{x}_0\text{''}} + \underbrace{\sqrt{1 - \alpha_{t-1} - \sigma_t^2} \cdot \epsilon_\theta^{(t)}(\boldsymbol{x}_t)}_{\text{``direction pointing to } \boldsymbol{x}_t\text{''}} + \underbrace{\sigma_t \epsilon_t}_{\text{random noise}} \tag{12}$$

where $\epsilon_t \sim \mathcal{N}(\boldsymbol{0}, \boldsymbol{I})$ is standard Gaussian noise independent of $\boldsymbol{x}_t$, and we define $\alpha_0 := 1$. Different choices of $\sigma$ values results in different generative processes, all while using the same model $\epsilon_\theta$, so re-training the model is unnecessary. When $\sigma_t = \sqrt{(1 - \alpha_{t-1})/(1 - \alpha_t)}\sqrt{1 - \alpha_t/\alpha_{t-1}}$ for all $t$, the forward process becomes Markovian, and the generative process becomes a DDPM.

We note another special case when $\sigma_t = 0$ for all $t$[5]; the forward process becomes deterministic given $\boldsymbol{x}_{t-1}$ and $\boldsymbol{x}_0$, except for $t = 1$; in the generative process, the coefficient before the random noise $\epsilon_t$ becomes zero. The resulting model becomes an implicit probabilistic model (Mohamed & Lakshminarayanan, 2016), where samples are generated from latent variables with a fixed procedure (from $\boldsymbol{x}_T$ to $\boldsymbol{x}_0$). We name this the *denoising diffusion implicit model* (DDIM, pronounced /d:ɪm/), because it is an implicit probabilistic model trained with the DDPM objective (despite the forward process no longer being a diffusion).

## 4.2 ACCELERATED GENERATION PROCESSES

In the previous sections, the generative process is considered as the approximation to the reverse process; since of the forward process has $T$ steps, the generative process is also forced to sample $T$ steps. However, as the denoising objective $L_{\mathbf{1}}$ does not depend on the specific forward procedure as long as $q_\sigma(\boldsymbol{x}_t|\boldsymbol{x}_0)$ is fixed, we may also consider forward processes with lengths smaller than $T$, which accelerates the corresponding generative processes without having to train a different model.

Let us consider the forward process as defined not on all the latent variables $\boldsymbol{x}_{1:T}$, but on a subset $\{\boldsymbol{x}_{\tau_1}, \ldots, \boldsymbol{x}_{\tau_S}\}$, where $\tau$ is an increasing sub-sequence of $[1, \ldots, T]$ of length $S$. In particular, we define the sequential forward process over $\boldsymbol{x}_{\tau_1}, \ldots, \boldsymbol{x}_{\tau_S}$ such that $q(\boldsymbol{x}_{\tau_i}|\boldsymbol{x}_0) = \mathcal{N}(\sqrt{\alpha_{\tau_i}}\boldsymbol{x}_0, (1 - \alpha_{\tau_i})\boldsymbol{I})$ matches the "marginals" (see Figure 2 for an illustration). The generative process now samples latent variables according to reversed($\tau$), which we term *(sampling) trajectory*. When the length of the sampling trajectory is much smaller than $T$, we may achieve significant increases in computational efficiency due to the iterative nature of the sampling process.

Using a similar argument as in Section 3, we can justify using the model trained with the $L_{\mathbf{1}}$ objective, so no changes are needed in training. We show that only slight changes to the updates in Eq. (12) are needed to obtain the new, faster generative processes, which applies to DDPM, DDIM, as well as all generative processes considered in Eq. (10). We include these details in Appendix C.1.

In principle, this means that we can train a model with an arbitrary number of forward steps but only sample from some of them in the generative process. Therefore, the trained model could consider many more steps than what is considered in (Ho et al., 2020) or even a continuous time variable $t$ (Chen et al., 2020). We leave empirical investigations of this aspect as future work.

---

[5]Although this case is not covered in Theorem 1, we can always approximate it by making $\sigma_t$ very small.

### 4.3 RELEVANCE TO NEURAL ODES

Moreover, we can rewrite the DDIM iterate according to Eq. (12), and its similarity to Euler integration for solving ODEs becomes more apparent:

$$\sqrt{\frac{1}{\alpha_{t-1}}}\boldsymbol{x}_{t-1} = \sqrt{\frac{1}{\alpha_t}}\boldsymbol{x}_t + \left(\sqrt{\frac{1-\alpha_{t-1}}{\alpha_{t-1}}} - \sqrt{\frac{1-\alpha_t}{\alpha_t}}\right)\epsilon_\theta^{(t)}(\boldsymbol{x}_t) \tag{13}$$

We can reparameterize $(\sqrt{1-\alpha}/\sqrt{\alpha})$ with $\lambda$ and $(\boldsymbol{x}/\sqrt{\alpha})$ with $H(\lambda)$ then sampling $\boldsymbol{x}_0$ with Equation (13) can be treated as integration over the following ODE:

$$H(0) = \int_M^0 \epsilon_\theta^\lambda(H(\lambda)\sqrt{\lambda^2-1})\mathrm{d}\lambda + H(M), \quad H(M) \sim \mathcal{N}(\boldsymbol{0}, \boldsymbol{I}) \tag{14}$$

for some very large $M$ (which corresponds to the case of $\alpha \approx 0$). This suggests that with enough $T$ (discretization steps), the we can also reverse the generation process (going from $t = 0$ to $T$), which encodes $\boldsymbol{x}_0$ to $\boldsymbol{x}_T$ and simulates the reverse of the ODE in Eq. (14). This suggests that unlike DDPM, we can use DDIM to obtain encodings of the observations (as the form of $\boldsymbol{x}_T$), which might be useful for other downstream applications that requires latent representations of a model.

## 5 EXPERIMENTS

In this section, we show that DDIMs outperform DDPMs in terms of image generation when fewer iterations are considered, giving speed ups of $10\times$ to $100\times$ over the original DDPM generation process. Moreover, unlike DDPMs, once the initial latent variables $\boldsymbol{x}_T$ are fixed, DDIMs retain high-level image features regardless of the generation trajectory, so they are able to perform interpolation directly from the latent space. DDIMs can also be used to encode samples that reconstruct them from the latent code, which DDPMs cannot do due to the stochastic sampling process.

For each dataset, we use the **same trained model** with $T = 1000$ and the objective being $L_\gamma$ from Eq. (5) with $\gamma = \boldsymbol{1}$; as we argued in Section 3, no changes are needed with regards to the training procedure. The only changes that we make is **how we produce samples from the model**; we achieve this by controlling $\tau$ (which controls how fast the samples are obtained) and $\sigma$ (which interpolates between the deterministic DDIM and the stochastic DDPM).

We consider different sub-sequences $\tau$ of $[1, \dots, T]$ and different variance hyperparameters $\sigma$ indexed by elements of $\tau$. To simplify comparisons, we consider $\sigma$ with the form:

$$\sigma_{\tau_i}(\eta) = \eta\sqrt{(1-\alpha_{\tau_{i-1}})/(1-\alpha_{\tau_i})}\sqrt{1-\alpha_{\tau_i}/\alpha_{\tau_{i-1}}}, \tag{15}$$

where $\eta \in \mathbb{R}_{\geq 0}$ is a hyperparameter that we can directly control. This includes an original DDPM generative process when $\eta = 1$ and DDIM when $\eta = 0$. We also consider DDPM where the random noise has a larger standard deviation than $\sigma(1)$, which we denote as $\hat{\sigma}$: $\hat{\sigma}_{\tau_i} = \sqrt{1-\alpha_{\tau_i}/\alpha_{\tau_{i-1}}}$. This is used by the implementation in Ho et al. (2020) **only to obtain the CIFAR10 samples**, but not samples of the other datasets. We include more details in Appendix D.

### 5.1 SAMPLE QUALITY AND EFFICIENCY

In Table 1, we report the quality of the generated samples with models trained on CIFAR10 and CelebA, as measured by Frechet Inception Distance (FID (Heusel et al., 2017)), where we vary the number of timesteps used to generate a sample $(\dim(\tau))$ and the stochasticity of the process $(\eta)$. As expected, the sample quality becomes higher as we increase $\dim(\tau)$, presenting a trade-off between sample quality and computational costs. We observe that DDIM $(\eta = 0)$ achieves the best sample quality when $\dim(\tau)$ is small, and DDPM $(\eta = 1$ and $\hat{\sigma})$ typically has worse sample quality compared to its less stochastic counterparts with the same $\dim(\tau)$, except for the case for $\dim(\tau) = 1000$ and $\hat{\sigma}$ reported by Ho et al. (2020) where DDIM is marginally worse. However, the sample quality of $\hat{\sigma}$ becomes much worse for smaller $\dim(\tau)$, which suggests that it is ill-suited for shorter trajectories. DDIM, on the other hand, achieves high sample quality much more consistently.

In Figure 3, we show CIFAR10 and CelebA samples with the same number of sampling steps and varying $\sigma$. For the DDPM, the sample quality deteriorates rapidly when the sampling trajectory has

Table 1: CIFAR10 and CelebA image generation measured in FID. $\eta = 1.0$ and $\hat{\sigma}$ are cases of DDPM (although Ho et al. (2020) only considered $T = 1000$ steps, and $S < T$ can be seen as simulating DDPMs trained with $S$ steps), and $\eta = 0.0$ indicates DDIM.

|   | $S$ | CIFAR10 ($32 \times 32$) | | | | | CelebA ($64 \times 64$) | | | | |
|---|---|---|---|---|---|---|---|---|---|---|---|
|   |   | 10 | 20 | 50 | 100 | 1000 | 10 | 20 | 50 | 100 | 1000 |
|   | 0.0 | **13.36** | **6.84** | **4.67** | **4.16** | 4.04 | **17.33** | **13.73** | **9.17** | **6.53** | 3.51 |
|   | 0.2 | 14.04 | 7.11 | 4.77 | 4.25 | 4.09 | 17.66 | 14.11 | 9.51 | 6.79 | 3.64 |
| $\eta$ | 0.5 | 16.66 | 8.35 | 5.25 | 4.46 | 4.29 | 19.86 | 16.06 | 11.01 | 8.09 | 4.28 |
|   | 1.0 | 41.07 | 18.36 | 8.01 | 5.78 | 4.73 | 33.12 | 26.03 | 18.48 | 13.93 | 5.98 |
|   | $\hat{\sigma}$ | 367.43 | 133.37 | 32.72 | 9.99 | **3.17** | 299.71 | 183.83 | 71.71 | 45.20 | **3.26** |



Figure 3: CIFAR10 and CelebA samples with $\dim(\tau) = 10$ and $\dim(\tau) = 100$.

10 steps. For the case of $\hat{\sigma}$, the generated images seem to have more noisy perturbations under short trajectories; this explains why the FID scores are much worse than other methods, as FID is very sensitive to such perturbations (as discussed in Jolicoeur-Martineau et al. (2020)).

In Figure 4, we show that the amount of time needed to produce a sample scales linearly with the length of the sample trajectory. This suggests that DDIM is useful for producing samples more efficiently, as samples can be generated in much fewer steps. Notably, DDIM is able to produce samples with quality comparable to 1000 step models within 20 to 100 steps, which is a $10\times$ to $50\times$ speed up compared to the original DDPM. Even though DDPM could also achieve reasonable sample quality with $100\times$ steps, DDIM requires much fewer steps to achieve this; on CelebA, the FID score of the 100 step DDPM is similar to that of the 20 step DDIM.

## 5.2 SAMPLE CONSISTENCY IN DDIMS

For DDIM, the generative process is deterministic, and $\boldsymbol{x}_0$ would depend only on the initial state $\boldsymbol{x}_T$. In Figure 5, we observe the generated images under different generative trajectories (i.e. different $\tau$) while starting with the same initial $\boldsymbol{x}_T$. Interestingly, for the generated images with the same initial $\boldsymbol{x}_T$, most high-level features are similar, regardless of the generative trajectory. In many cases, samples generated with only 20 steps are already very similar to ones generated with 1000 steps in terms of high-level features, with only minor differences in details. Therefore, it would appear that $\boldsymbol{x}_T$ alone would be an informative latent encoding of the image; and minor details that affects sample quality are encoded in the parameters, as longer sample trajectories gives better quality samples but do not significantly affect the high-level features. We show more samples in Appendix D.4.

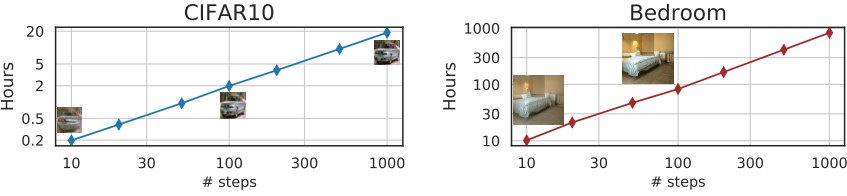

Figure 4: Hours to sample 50k images with one Nvidia 2080 Ti GPU and samples at different steps.

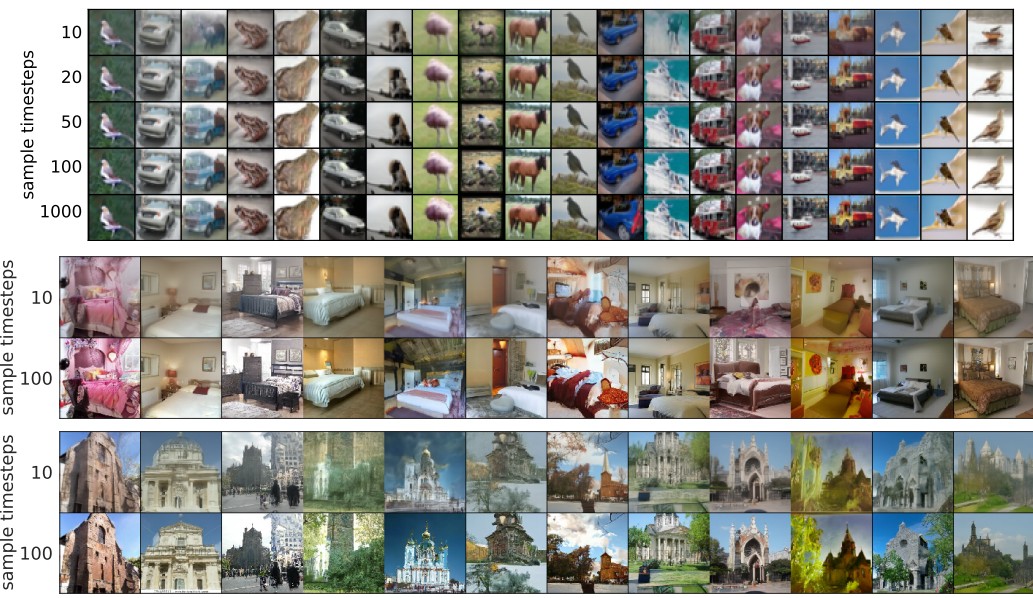

Figure 5: Samples from DDIM with the same random $\boldsymbol{x}_T$ and different number of steps.

## 5.3 Interpolation in deterministic generative processes

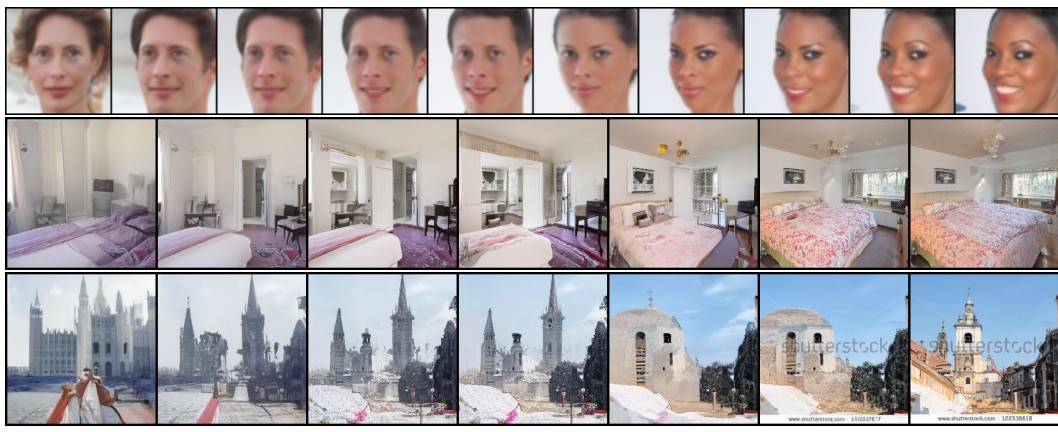

Figure 6: Interpolation of samples from DDIM with $\dim(\tau) = 50$.

Since the high level features of the DDIM sample is encoded by $\boldsymbol{x}_T$, we are interested to see whether it would exhibit the semantic interpolation effect similar to that observed in other implicit probabilistic models, such as GANs (Goodfellow et al., 2014). This is different from the interpolation procedure in Ho et al. (2020), since in DDPM the same $\boldsymbol{x}_T$ would lead to highly diverse $\boldsymbol{x}_0$ due to the stochastic generative process[6]. In Figure 6, we show that simple interpolations in $\boldsymbol{x}_T$ can lead to semantically meaningful interpolations between two samples. We include more details and samples in Appendix D.5. This allows DDIM to control the generated images on a high level directly through the latent variables, which DDPMs cannot.

## 5.4 Reconstruction from Latent Space

As DDIM is the Euler integration for a particular ODE, it would be interesting to see whether it can encode from $\boldsymbol{x}_0$ to $\boldsymbol{x}_T$ (reverse of Eq. (14)) and reconstruct $\boldsymbol{x}_0$ from the resulting $\boldsymbol{x}_T$ (forward

---

[6]Although it might be possible if one interpolates all $T$ noises, like what is done in Song & Ermon (2020).

Table 2: Reconstruction error with DDIM on CIFAR-10 test set, rounded to $10^{-4}$.

| $S$ | 10 | 20 | 50 | 100 | 200 | 500 | 1000 |
|---|---|---|---|---|---|---|---|
| Error | 0.014 | 0.0065 | 0.0023 | 0.0009 | 0.0004 | 0.0001 | 0.0001 |

of Eq. (14))[7]. We consider encoding and decoding on the CIFAR-10 test set with the CIFAR-10 model with $S$ steps for both encoding and decoding; we report the per-dimension mean squared error (scaled to $[0, 1]$) in Table 2. Our results show that DDIMs have lower reconstruction error for larger $S$ values and have properties similar to Neural ODEs and normalizing flows. The same cannot be said for DDPMs due to their stochastic nature.

## 6 RELATED WORK

Our work is based on a large family of existing methods on learning generative models as transition operators of Markov chains (Sohl-Dickstein et al., 2015; Bengio et al., 2014; Salimans et al., 2014; Song et al., 2017; Goyal et al., 2017; Levy et al., 2017). Among them, denoising diffusion probabilistic models (DDPMs, Ho et al. (2020)) and noise conditional score networks (NCSN, Song & Ermon (2019; 2020)) have recently achieved high sample quality comparable to GANs (Brock et al., 2018; Karras et al., 2018). DDPMs optimize a variational lower bound to the log-likelihood, whereas NCSNs optimize the score matching objective (Hyvärinen, 2005) over a nonparametric Parzen density estimator of the data (Vincent, 2011; Raphan & Simoncelli, 2011).

Despite their different motivations, DDPMs and NCSNs are closely related. Both use a denoising autoencoder objective for many noise levels, and both use a procedure similar to Langevin dynamics to produce samples (Neal et al., 2011). Since Langevin dynamics is a discretization of a gradient flow (Jordan et al., 1998), both DDPM and NCSN require many steps to achieve good sample quality. This aligns with the observation that DDPM and existing NCSN methods have trouble generating high-quality samples in a few iterations.

DDIM, on the other hand, is an implicit generative model (Mohamed & Lakshminarayanan, 2016) where samples are uniquely determined from the latent variables. Hence, DDIM has certain properties that resemble GANs (Goodfellow et al., 2014) and invertible flows (Dinh et al., 2016), such as the ability to produce semantically meaningful interpolations. We derive DDIM from a purely variational perspective, where the restrictions of Langevin dynamics are not relevant; this could partially explain why we are able to observe superior sample quality compared to DDPM under fewer iterations. The sampling procedure of DDIM is also reminiscent of neural networks with continuous depth (Chen et al., 2018; Grathwohl et al., 2018), since the samples it produces from the same latent variable have similar high-level visual features, regardless of the specific sample trajectory.

## 7 DISCUSSION

We have presented DDIMs – an implicit generative model trained with denoising auto-encoding / score matching objectives – from a purely variational perspective. DDIM is able to generate high-quality samples much more efficiently than existing DDPMs and NCSNs, with the ability to perform meaningful interpolations from the latent space. The non-Markovian forward process presented here seems to suggest continuous forward processes other than Gaussian (which cannot be done in the original diffusion framework, since Gaussian is the only stable distribution with finite variance). We also demonstrated a discrete case with a multinomial forward process in Appendix A, and it would be interesting to investigate similar alternatives for other combinatorial structures.

Moreover, since the sampling procedure of DDIMs is similar to that of an neural ODE, it would be interesting to see if methods that decrease the discretization error in ODEs, including multi-step methods such as Adams-Bashforth (Butcher & Goodwin, 2008), could be helpful for further improving sample quality in fewer steps (Queiruga et al., 2020). It is also relevant to investigate whether DDIMs exhibit other properties of existing implicit models (Bau et al., 2019).

---

[7]Since $x_T$ and $x_0$ have the same dimensions, their compression qualities are not our immediate concern.

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

## A    NON-MARKOVIAN FORWARD PROCESSES FOR A DISCRETE CASE

In this section, we describe a non-Markovian forward processes for discrete data and corresponding variational objectives. Since the focus of this paper is to accelerate reverse models corresponding to the Gaussian diffusion, we leave empirical evaluations as future work.

For a categorical observation $\boldsymbol{x}_0$ that is a one-hot vector with $K$ possible values, we define the forward process as follows. First, we have $q(\boldsymbol{x}_t|\boldsymbol{x}_0)$ as the following categorical distribution:

$$q(\boldsymbol{x}_t|\boldsymbol{x}_0) = \mathrm{Cat}(\alpha_t \boldsymbol{x}_0 + (1 - \alpha_t)\mathbf{1}_K) \tag{16}$$

where $\mathbf{1}_K \in \mathbb{R}^K$ is a vector with all entries being $1/K$, and $\alpha_t$ decreasing from $\alpha_0 = 1$ for $t = 0$ to $\alpha_T = 0$ for $t = T$. Then we define $q(\boldsymbol{x}_{t-1}|\boldsymbol{x}_t, \boldsymbol{x}_0)$ as the following mixture distribution:

$$q(\boldsymbol{x}_{t-1}|\boldsymbol{x}_t, \boldsymbol{x}_0) = \begin{cases} \mathrm{Cat}(\boldsymbol{x}_t) & \text{with probability } \sigma_t \\ \mathrm{Cat}(\boldsymbol{x}_0) & \text{with probability } (\alpha_{t-1} - \sigma_t \alpha_t) \\ \mathrm{Cat}(\mathbf{1}_K) & \text{with probability } (1 - \alpha_{t-1}) - (1 - \alpha_t)\sigma_t \end{cases}, \tag{17}$$

or equivalently:

$$q(\boldsymbol{x}_{t-1}|\boldsymbol{x}_t, \boldsymbol{x}_0) = \mathrm{Cat}\left(\sigma_t \boldsymbol{x}_t + (\alpha_{t-1} - \sigma_t \alpha_t)\boldsymbol{x}_0 + ((1 - \alpha_{t-1}) - (1 - \alpha_t)\sigma_t)\mathbf{1}_K\right), \tag{18}$$

which is consistent with how we have defined $q(\boldsymbol{x}_t|\boldsymbol{x}_0)$.

Similarly, we can define our reverse process $p_\theta(\boldsymbol{x}_{t-1}|\boldsymbol{x}_t)$ as:

$$p_\theta(\boldsymbol{x}_{t-1}|\boldsymbol{x}_t) = \mathrm{Cat}\left(\sigma_t \boldsymbol{x}_t + (\alpha_{t-1} - \sigma_t \alpha_t)f_\theta^{(t)}(\boldsymbol{x}_t) + ((1 - \alpha_{t-1}) - (1 - \alpha_t)\sigma_t)\mathbf{1}_K\right), \tag{19}$$

where $f_\theta^{(t)}(\boldsymbol{x}_t)$ maps $\boldsymbol{x}_t$ to a $K$-dimensional vector. As $(1 - \alpha_{t-1}) - (1 - \alpha_t)\sigma_t \to 0$, the sampling process will become less stochastic, in the sense that it will either choose $\boldsymbol{x}_t$ or the predicted $\boldsymbol{x}_0$ with high probability. The KL divergence

$$D_{\mathrm{KL}}(q(\boldsymbol{x}_{t-1}|\boldsymbol{x}_t, \boldsymbol{x}_0)\|p_\theta(\boldsymbol{x}_{t-1}|\boldsymbol{x}_t)) \tag{20}$$

is well-defined, and is simply the KL divergence between two categoricals. Therefore, the resulting variational objective function should be easy to optimize as well. Moreover, as KL divergence is convex, we have this upper bound (which is tight when the right hand side goes to zero):

$$D_{\mathrm{KL}}(q(\boldsymbol{x}_{t-1}|\boldsymbol{x}_t, \boldsymbol{x}_0)\|p_\theta(\boldsymbol{x}_{t-1}|\boldsymbol{x}_t)) \leq (\alpha_{t-1} - \sigma_t \alpha_t)D_{\mathrm{KL}}(\mathrm{Cat}(\boldsymbol{x}_0)\|\mathrm{Cat}(f_\theta^{(t)}(\boldsymbol{x}_t))).$$

The right hand side is simply a multi-class classification loss (up to constants), so we can arrive at similar arguments regarding how changes in $\sigma_t$ do not affect the objective (up to re-weighting).

## B    PROOFS

**Lemma 1.** *For $q_\sigma(\boldsymbol{x}_{1:T}|\boldsymbol{x}_0)$ defined in Eq. (6) and $q_\sigma(\boldsymbol{x}_{t-1}|\boldsymbol{x}_t, \boldsymbol{x}_0)$ defined in Eq. (7), we have:*

$$q_\sigma(\boldsymbol{x}_t|\boldsymbol{x}_0) = \mathcal{N}(\sqrt{\alpha_t}\boldsymbol{x}_0, \sqrt{1 - \alpha_t}\boldsymbol{I}) \tag{21}$$

*Proof.* Assume for any $t \leq T$, $q_\sigma(\boldsymbol{x}_t|\boldsymbol{x}_0) = \mathcal{N}(\sqrt{\alpha_t}\boldsymbol{x}_0, \sqrt{1 - \alpha_t}\boldsymbol{I})$ holds, if:

$$q_\sigma(\boldsymbol{x}_{t-1}|\boldsymbol{x}_0) = \mathcal{N}(\sqrt{\alpha_{t-1}}\boldsymbol{x}_0, \sqrt{1 - \alpha_{t-1}}\boldsymbol{I}) \tag{22}$$

then we can prove the statement with an induction argument for $t$ from $T$ to 1, since the base case $(t = T)$ already holds.

First, we have that

$$q_\sigma(\boldsymbol{x}_{t-1}|\boldsymbol{x}_0) := \int_{\boldsymbol{x}_t} q_\sigma(\boldsymbol{x}_t|\boldsymbol{x}_0)q_\sigma(\boldsymbol{x}_{t-1}|\boldsymbol{x}_t, \boldsymbol{x}_0)\mathrm{d}\boldsymbol{x}_t$$

and

$$q_\sigma(\boldsymbol{x}_t|\boldsymbol{x}_0) = \mathcal{N}(\sqrt{\alpha_t}\boldsymbol{x}_0, (1 - \alpha_t)\boldsymbol{I}) \tag{23}$$

$$q_\sigma(\boldsymbol{x}_{t-1}|\boldsymbol{x}_t, \boldsymbol{x}_0) = \mathcal{N}\left(\sqrt{\alpha_{t-1}}\boldsymbol{x}_0 + \sqrt{1 - \alpha_{t-1} - \sigma_t^2} \cdot \frac{\boldsymbol{x}_t - \sqrt{\alpha_t}\boldsymbol{x}_0}{\sqrt{1 - \alpha_t}}, \sigma_t^2 \boldsymbol{I}\right). \tag{24}$$

From Bishop (2006) (2.115), we have that $\hat{q}(\boldsymbol{x}_{t-1}|\boldsymbol{x}_0)$ is Gaussian and

$$\mathbb{E}[q_\sigma(\boldsymbol{x}_{t-1}|\boldsymbol{x}_0)] = \sqrt{\alpha_{t-1}}\boldsymbol{x}_0 + \sqrt{1 - \alpha_{t-1} - \sigma_t^2} \cdot \frac{\sqrt{\alpha_t}\boldsymbol{x}_0 - \sqrt{\alpha_t}\boldsymbol{x}_0}{\sqrt{1 - \alpha_t}} \tag{25}$$

$$= \sqrt{\alpha_{t-1}}\boldsymbol{x}_0 \tag{26}$$

and

$$\mathrm{Cov}[q_\sigma(\boldsymbol{x}_{t-1}|\boldsymbol{x}_0)] = \sigma_t^2\boldsymbol{I} + \frac{1 - \alpha_{t-1} - \sigma_t^2}{1 - \alpha_t}(1 - \alpha_t)\boldsymbol{I} = (1 - \alpha_{t-1})\boldsymbol{I} \tag{27}$$

Therefore, $q_\sigma(\boldsymbol{x}_{t-1}|\boldsymbol{x}_0) = \mathcal{N}(\sqrt{\alpha_{t-1}}\boldsymbol{x}_0, \sqrt{1 - \alpha_{t-1}}\boldsymbol{I})$, which allows us to apply the induction argument. $\square$

**Theorem 1.** *For all $\sigma > 0$, there exists $\gamma \in \mathbb{R}_{>0}^T$ and $C \in \mathbb{R}$, such that $J_\sigma = L_\gamma + C$.*

*Proof.* From the definition of $J_\sigma$:

$$J_\sigma(\epsilon_\theta) := \mathbb{E}_{\boldsymbol{x}_{0:T}\sim q(\boldsymbol{x}_{0:T})}\left[q_\sigma(\boldsymbol{x}_T|\boldsymbol{x}_0) + \sum_{t=2}^T \log q_\sigma(\boldsymbol{x}_{t-1}|\boldsymbol{x}_t, \boldsymbol{x}_0) - \sum_{t=1}^T \log p_\theta^{(t)}(\boldsymbol{x}_{t-1}|\boldsymbol{x}_t)\right] \tag{28}$$

$$\equiv \mathbb{E}_{\boldsymbol{x}_{0:T}\sim q(\boldsymbol{x}_{0:T})}\left[\sum_{t=2}^T D_{\mathrm{KL}}(q_\theta(\boldsymbol{x}_{t-1}|\boldsymbol{x}_t, \boldsymbol{x}_0))\|p_\theta^{(t)}(\boldsymbol{x}_{t-1}|\boldsymbol{x}_t)) - \log p_\theta^{(1)}(\boldsymbol{x}_0|\boldsymbol{x}_1)\right]$$

where we use $\equiv$ to denote "equal up to a value that does not depend on $\epsilon_\theta$ (but may depend on $q_\sigma$)". For $t > 1$:

$$\mathbb{E}_{\boldsymbol{x}_0, \boldsymbol{x}_t\sim q(\boldsymbol{x}_0, \boldsymbol{x}_t)}[D_{\mathrm{KL}}(q_\sigma(\boldsymbol{x}_{t-1}|\boldsymbol{x}_t, \boldsymbol{x}_0))\|p_\theta^{(t)}(\boldsymbol{x}_{t-1}|\boldsymbol{x}_t))]$$

$$= \mathbb{E}_{\boldsymbol{x}_0, \boldsymbol{x}_t\sim q(\boldsymbol{x}_0, \boldsymbol{x}_t)}[D_{\mathrm{KL}}(q_\sigma(\boldsymbol{x}_{t-1}|\boldsymbol{x}_t, \boldsymbol{x}_0))\|q_\sigma(\boldsymbol{x}_{t-1}|\boldsymbol{x}_t, f_\theta^{(t)}(\boldsymbol{x}_t)))]$$

$$= \mathbb{E}_{\boldsymbol{x}_0, \boldsymbol{x}_t\sim q(\boldsymbol{x}_0, \boldsymbol{x}_t)}\left[\frac{\|\boldsymbol{x}_0 - f_\theta^{(t)}(\boldsymbol{x}_t)\|_2^2}{2\sigma_t^2}\right] \tag{29}$$

$$= \mathbb{E}_{\boldsymbol{x}_0\sim q(\boldsymbol{x}_0), \epsilon\sim\mathcal{N}(\boldsymbol{0}, \boldsymbol{I}), \boldsymbol{x}_t=\sqrt{\alpha_t}\boldsymbol{x}_0 + \sqrt{1-\alpha_t}\epsilon}\left[\frac{\|(\boldsymbol{x}_t - \epsilon)/\sqrt{\alpha_t} - (\boldsymbol{x}_t - \epsilon_\theta^{(t)}(\boldsymbol{x}_t))/\sqrt{\alpha_t}\|_2^2}{2\sigma_t^2}\right] \tag{30}$$

$$= \mathbb{E}_{\boldsymbol{x}_0\sim q(\boldsymbol{x}_0), \epsilon\sim\mathcal{N}(\boldsymbol{0}, \boldsymbol{I}), \boldsymbol{x}_t=\sqrt{\alpha_t}\boldsymbol{x}_0 + \sqrt{1-\alpha_t}\epsilon}\left[\frac{\|\epsilon - \epsilon_\theta^{(t)}(\boldsymbol{x}_t)\|_2^2}{2d\sigma_t^2\alpha_t}\right] \tag{31}$$

where $d$ is the dimension of $\boldsymbol{x}_0$. For $t = 0$:

$$\mathbb{E}_{\boldsymbol{x}_0, \boldsymbol{x}_1\sim q(\boldsymbol{x}_0, \boldsymbol{x}_1)}\left[-\log p_\theta^{(1)}(\boldsymbol{x}_0|\boldsymbol{x}_1)\right] \equiv \mathbb{E}_{\boldsymbol{x}_0, \boldsymbol{x}_1\sim q(\boldsymbol{x}_0, \boldsymbol{x}_1)}\left[\frac{\|\boldsymbol{x}_0 - f_\theta^{(t)}(\boldsymbol{x}_1)\|_2^2}{2\sigma_1^2}\right] \tag{32}$$

$$= \mathbb{E}_{\boldsymbol{x}_0\sim q(\boldsymbol{x}_0), \epsilon\sim\mathcal{N}(\boldsymbol{0}, \boldsymbol{I}), \boldsymbol{x}_1=\sqrt{\alpha_1}\boldsymbol{x}_0 + \sqrt{1-\alpha_t}\epsilon}\left[\frac{\|\epsilon - \epsilon_\theta^{(1)}(\boldsymbol{x}_1)\|_2^2}{2d\sigma_1^2\alpha_1}\right] \tag{33}$$

Therefore, when $\gamma_t = 1/(2d\sigma_t^2\alpha_t)$ for all $t \in \{1, \ldots, T\}$, we have

$$J_\sigma(\epsilon_\theta) \equiv \sum_{t=1}^T \frac{1}{2d\sigma_t^2\alpha_t}\mathbb{E}\left[\|\epsilon_\theta^{(t)}(\boldsymbol{x}_t) - \epsilon_t\|_2^2\right] = L_\gamma(\epsilon_\theta) \tag{34}$$

for all $\epsilon_\theta$. From the definition of "$\equiv$", we have that $J_\sigma = L_\gamma + C$. $\square$

## C  ADDITIONAL DERIVATIONS

### C.1  ACCELERATED SAMPLING PROCESSES

In the accelerated case, we can consider the inference process to be factored as:

$$q_{\sigma,\tau}(\boldsymbol{x}_{1:T}|\boldsymbol{x}_0) = q_{\sigma,\tau}(\boldsymbol{x}_{\tau_S}|\boldsymbol{x}_0)\prod_{i=1}^{S}q_{\sigma,\tau}(\boldsymbol{x}_{\tau_{i-1}}|\boldsymbol{x}_{\tau_i},\boldsymbol{x}_0)\prod_{t\in\bar{\tau}}q_{\sigma,\tau}(\boldsymbol{x}_t|\boldsymbol{x}_0) \tag{35}$$

where $\tau$ is a sub-sequence of $[1,\ldots,T]$ of length $S$ with $\tau_S = T$, and let $\bar{\tau} := \{1,\ldots,T\}\setminus\tau$ be its complement. Intuitively, the graphical model of $\{\boldsymbol{x}_{\tau_i}\}_{i=1}^{S}$ and $\boldsymbol{x}_0$ form a chain, whereas the graphical model of $\{\boldsymbol{x}_t\}_{t\in\bar{\tau}}$ and $\boldsymbol{x}_0$ forms a star graph. We define:

$$q_{\sigma,\tau}(\boldsymbol{x}_t|\boldsymbol{x}_0) = \mathcal{N}(\sqrt{\alpha_t}\boldsymbol{x}_0,(1-\alpha_t)\boldsymbol{I}) \quad \forall t\in\bar{\tau}\cup\{T\} \tag{36}$$

$$q_{\sigma,\tau}(\boldsymbol{x}_{\tau_{i-1}}|\boldsymbol{x}_{\tau_i},\boldsymbol{x}_0) = \mathcal{N}\left(\sqrt{\alpha_{\tau_{i-1}}}\boldsymbol{x}_0 + \sqrt{1-\alpha_{\tau_{i-1}}-\sigma_{\tau_i}^2}\cdot\frac{\boldsymbol{x}_{\tau_i}-\sqrt{\alpha_{\tau_i}}\boldsymbol{x}_0}{\sqrt{1-\alpha_{\tau_i}}},\sigma_{\tau_i}^2\boldsymbol{I}\right) \quad \forall i\in[S]$$

where the coefficients are chosen such that:

$$q_{\sigma,\tau}(\boldsymbol{x}_{\tau_i}|\boldsymbol{x}_0) = \mathcal{N}(\sqrt{\alpha_{\tau_i}}\boldsymbol{x}_0,(1-\alpha_{\tau_i})\boldsymbol{I}) \quad \forall i\in[S] \tag{37}$$

i.e., the "marginals" match.

The corresponding "generative process" is defined as:

$$p_\theta(\boldsymbol{x}_{0:T}) := p_\theta(\boldsymbol{x}_T)\underbrace{\prod_{i=1}^{S}p_\theta^{(\tau_i)}(\boldsymbol{x}_{\tau_{i-1}}|\boldsymbol{x}_{\tau_i})}_{\text{use to produce samples}}\times\underbrace{\prod_{t\in\bar{\tau}}p_\theta^{(t)}(\boldsymbol{x}_0|\boldsymbol{x}_t)}_{\text{in variational objective}} \tag{38}$$

where only part of the models are actually being used to produce samples. The conditionals are:

$$p_\theta^{(\tau_i)}(\boldsymbol{x}_{\tau_{i-1}}|\boldsymbol{x}_{\tau_i}) = q_{\sigma,\tau}(\boldsymbol{x}_{\tau_i}|\boldsymbol{x}_{\tau_i},f_\theta^{(\tau_i)}(\boldsymbol{x}_{\tau_{i-1}})) \quad \text{if } i\in[S], i>1 \tag{39}$$

$$p_\theta^{(t)}(\boldsymbol{x}_0|\boldsymbol{x}_t) = \mathcal{N}(f_\theta^{(t)}(\boldsymbol{x}_t),\sigma_t^2\boldsymbol{I}) \quad \text{otherwise,} \tag{40}$$

where we leverage $q_{\sigma,\tau}(\boldsymbol{x}_{\tau_{i-1}}|\boldsymbol{x}_{\tau_i},\boldsymbol{x}_0)$ as part of the inference process (similar to what we have done in Section 3). The resulting variational objective becomes (define $\boldsymbol{x}_{\tau_{L+1}}=\varnothing$ for conciseness):

$$J(\epsilon_\theta) = \mathbb{E}_{\boldsymbol{x}_{0:T}\sim q_{\sigma,\tau}(\boldsymbol{x}_{0:T})}[\log q_{\sigma,\tau}(\boldsymbol{x}_{1:T}|\boldsymbol{x}_0)-\log p_\theta(\boldsymbol{x}_{0:T})] \tag{41}$$

$$= \mathbb{E}_{\boldsymbol{x}_{0:T}\sim q_{\sigma,\tau}(\boldsymbol{x}_{0:T})}\left[\sum_{t\in\bar{\tau}}D_{\mathrm{KL}}(q_{\sigma,\tau}(\boldsymbol{x}_t|\boldsymbol{x}_0)\|p_\theta^{(t)}(\boldsymbol{x}_0|\boldsymbol{x}_t))\right. \tag{42}$$

$$\left.+\sum_{i=1}^{L}D_{\mathrm{KL}}(q_{\sigma,\tau}(\boldsymbol{x}_{\tau_{i-1}}|\boldsymbol{x}_{\tau_i},\boldsymbol{x}_0)\|p_\theta^{(\tau_i)}(\boldsymbol{x}_{\tau_{i-1}}|\boldsymbol{x}_{\tau_i})))\right]$$

where each KL divergence is between two Gaussians with variance independent of $\theta$. A similar argument to the proof used in Theorem 1 can show that the variational objective $J$ can also be converted to an objective of the form $L_\gamma$.

### C.2  DERIVATION OF DENOISING OBJECTIVES FOR DDPMS

We note that in Ho et al. (2020), a diffusion hyperparameter $\beta_t$[8] is first introduced, and then relevant variables $\alpha_t := 1 - \beta_t$ and $\bar{\alpha}_t = \prod_{t=1}^{T}\alpha_t$ are defined. In this paper, we have used the notation $\alpha_t$ to represent the variable $\bar{\alpha}_t$ in Ho et al. (2020) for three reasons. First, it makes it more clear that we only need to choose one set of hyperparameters, reducing possible cross-references of the derived variables. Second, it allows us to introduce the generalization as well as the acceleration case easier, because the inference process is no longer motivated by a diffusion. Third, there exists an isomorphism between $\alpha_{1:T}$ and $1,\ldots,T$, which is not the case for $\beta_t$.

---

[8]In this section we use teal to color notations used in Ho et al. (2020).

In this section, we use $\beta_t$ and $\alpha_t$ to be more consistent with the derivation in Ho et al. (2020), where

$$\alpha_t = \frac{\alpha_t}{\alpha_{t-1}} \tag{43}$$

$$\beta_t = 1 - \frac{\alpha_t}{\alpha_{t-1}} \tag{44}$$

can be uniquely determined from $\alpha_t$ (i.e. $\bar{\alpha}_t$).

First, from the diffusion forward process:

$$q(\boldsymbol{x}_{t-1}|\boldsymbol{x}_t, \boldsymbol{x}_0) = \mathcal{N}\left(\underbrace{\frac{\sqrt{\alpha_{t-1}}\beta_t}{1-\alpha_t}\boldsymbol{x}_0 + \frac{\alpha_t(1-\alpha_{t-1})}{1-\alpha_t}\boldsymbol{x}_t}_{\tilde{\mu}(\boldsymbol{x}_t, \boldsymbol{x}_0)}, \frac{1-\alpha_{t-1}}{1-\alpha_t}\beta_t \boldsymbol{I}\right)$$

Ho et al. (2020) considered a specific type of $p_\theta^{(t)}(\boldsymbol{x}_{t-1}|\boldsymbol{x}_t)$:

$$p_\theta^{(t)}(\boldsymbol{x}_{t-1}|\boldsymbol{x}_t) = \mathcal{N}\left(\mu_\theta(\boldsymbol{x}_t, t), \sigma_t \boldsymbol{I}\right) \tag{45}$$

which leads to the following variational objective:

$$L := \mathbb{E}_{\boldsymbol{x}_{0:T} \sim q(\boldsymbol{x}_{0:T})}\left[q(\boldsymbol{x}_T|\boldsymbol{x}_0) + \sum_{t=2}^{T} \log q(\boldsymbol{x}_{t-1}|\boldsymbol{x}_t, \boldsymbol{x}_0) - \sum_{t=1}^{T} \log p_\theta^{(t)}(\boldsymbol{x}_{t-1}|\boldsymbol{x}_t)\right] \tag{46}$$

$$\equiv \mathbb{E}_{\boldsymbol{x}_{0:T} \sim q(\boldsymbol{x}_{0:T})}\left[\sum_{t=2}^{T} \underbrace{D_{\mathrm{KL}}(q(\boldsymbol{x}_{t-1}|\boldsymbol{x}_t, \boldsymbol{x}_0))\|p_\theta^{(t)}(\boldsymbol{x}_{t-1}|\boldsymbol{x}_t))}_{L_{t-1}} - \log p_\theta^{(1)}(\boldsymbol{x}_0|\boldsymbol{x}_1)\right]$$

One can write:

$$L_{t-1} = \mathbb{E}_q\left[\frac{1}{2\sigma_t^2}\|\mu_\theta(\boldsymbol{x}_t, t) - \tilde{\mu}(\boldsymbol{x}_t, \boldsymbol{x}_0)\|_2^2\right] \tag{47}$$

Ho et al. (2020) chose the parametrization

$$\mu_\theta(\boldsymbol{x}_t, t) = \frac{1}{\sqrt{\alpha_t}}\left(\boldsymbol{x}_t - \frac{\beta_t}{\sqrt{1-\alpha_t}}\epsilon_\theta(\boldsymbol{x}_t, t)\right) \tag{48}$$

which can be simplified to:

$$L_{t-1} = \mathbb{E}_{\boldsymbol{x}_0, \epsilon}\left[\frac{\beta_t^2}{2\sigma_t^2(1-\alpha_t)\alpha_t}\|\epsilon - \epsilon_\theta(\sqrt{\alpha_t}\boldsymbol{x}_0 + \sqrt{1-\alpha_t}\epsilon, t)\|_2^2\right] \tag{49}$$

# D   EXPERIMENTAL DETAILS

## D.1   DATASETS AND ARCHITECTURES

We consider 4 image datasets with various resolutions: CIFAR10 ($32 \times 32$, unconditional), CelebA ($64 \times 64$), LSUN Bedroom ($256 \times 256$) and LSUN Church ($256 \times 256$). For all datasets, we set the hyperparameters $\alpha$ according to the heuristic in (Ho et al., 2020) to make the results directly comparable. We use the same model for each dataset, and only compare the performance of different generative processes. For CIFAR10, Bedroom and Church, we obtain the pretrained checkpoints from the original DDPM implementation; for CelebA, we trained our own model using the denoising objective $L_1$.

Our architecture for $\epsilon_\theta^{(t)}(\boldsymbol{x}_t)$ follows that in Ho et al. (2020), which is a U-Net (Ronneberger et al., 2015) based on a Wide ResNet (Zagoruyko & Komodakis, 2016). We use the pretrained models from Ho et al. (2020) for CIFAR10, Bedroom and Church, and train our own model for the CelebA $64 \times 64$ model (since a pretrained model is not provided). Our CelebA model has five feature map resolutions from $64 \times 64$ to $4 \times 4$, and we use the original CelebA dataset (not CelebA-HQ) using the pre-processing technique from the StyleGAN (Karras et al., 2018) repository.

Table 3: LSUN Bedroom and Church image generation results, measured in FID. For 1000 steps DDPM, the FIDs are 6.36 for Bedroom and 7.89 for Church.

| $\dim(\tau)$ | Bedroom ($256 \times 256$) | | | | Church ($256 \times 256$) | | | |
|---|---|---|---|---|---|---|---|---|
| | 10 | 20 | 50 | 100 | 10 | 20 | 50 | 100 |
| DDIM ($\eta = 0.0$) | **16.95** | **8.89** | **6.75** | **6.62** | **19.45** | **12.47** | **10.84** | 10.58 |
| DDPM ($\eta = 1.0$) | 42.78 | 22.77 | 10.81 | 6.81 | 51.56 | 23.37 | 11.16 | **8.27** |

## D.2 REVERSE PROCESS SUB-SEQUENCE SELECTION

We consider two types of selection procedure for $\tau$ given the desired $\dim(\tau) < T$:

- **Linear**: we select the timesteps such that $\tau_i = \lfloor ci \rfloor$ for some $c$;
- **Quadratic**: we select the timesteps such that $\tau_i = \lfloor ci^2 \rfloor$ for some $c$.

The constant value $c$ is selected such that $\tau_{-1}$ is close to $T$. We used *quadratic* for CIFAR10 and *linear* for the remaining datasets. These choices achieve slightly better FID than their alternatives in the respective datasets.

## D.3 CLOSED FORM EQUATIONS FOR EACH SAMPLING STEP

From the general sampling equation in Eq. (12), we have the following update equation:

$$\boldsymbol{x}_{\tau_{i-1}}(\eta) = \sqrt{\alpha_{\tau_{i-1}}} \left( \frac{\boldsymbol{x}_{\tau_i} - \sqrt{1 - \alpha_{\tau_i}} \epsilon_\theta^{(\tau_i)}(\boldsymbol{x}_{\tau_i})}{\sqrt{\alpha_{\tau_i}}} \right) + \sqrt{1 - \alpha_{\tau_{i-1}} - \sigma_{\tau_i}(\eta)^2} \cdot \epsilon_\theta^{(\tau_i)}(\tau_i) + \sigma_{\tau_i}(\eta)\epsilon$$

where

$$\sigma_{\tau_i}(\eta) = \eta \sqrt{\frac{1 - \alpha_{\tau_{i-1}}}{1 - \alpha_{\tau_i}}} \sqrt{1 - \frac{\alpha_{\tau_i}}{\alpha_{\tau_{i-1}}}}$$

For the case of $\hat{\sigma}$ (DDPM with a larger variance), the update equation becomes:

$$\boldsymbol{x}_{\tau_{i-1}} = \sqrt{\alpha_{\tau_{i-1}}} \left( \frac{\boldsymbol{x}_{\tau_i} - \sqrt{1 - \alpha_{\tau_i}} \epsilon_\theta^{(\tau_i)}(\boldsymbol{x}_{\tau_i})}{\sqrt{\alpha_{\tau_i}}} \right) + \sqrt{1 - \alpha_{\tau_{i-1}} - \sigma_{\tau_i}(1)^2} \cdot \epsilon_\theta^{(\tau_i)}(\tau_i) + \hat{\sigma}_{\tau_i}\epsilon$$

which uses a different coefficient for $\epsilon$ compared with the update for $\eta = 1$, but uses the same coefficient for the non-stochastic parts. This update is more stochastic than the update for $\eta = 1$, which explains why it achieves worse performance when $\dim(\tau)$ is small.

## D.4 SAMPLES AND CONSISTENCY

We show more samples in Figure 7 (CIFAR10), Figure 8 (CelebA), Figure 10 (Church) and consistency results of DDIM in Figure 9 (CelebA).

## D.5 INTERPOLATION

To generate interpolations on a line, we randomly sample two initial $\boldsymbol{x}_T$ values from the standard Gaussian, interpolate them with spherical linear interpolation (Shoemake, 1985), and then use the DDIM to obtain $\boldsymbol{x}_0$ samples.

$$\boldsymbol{x}_T^{(\alpha)} = \frac{\sin((1 - \alpha)\theta)}{\sin(\theta)} \boldsymbol{x}_T^{(0)} + \frac{\sin(\alpha\theta)}{\sin(\theta)} \boldsymbol{x}_T^{(1)} \tag{50}$$

where $\theta = \arccos\left( \frac{(\boldsymbol{x}_T^{(0)})^\top \boldsymbol{x}_T^{(1)}}{\|\boldsymbol{x}_T^{(0)}\|\|\boldsymbol{x}_T^{(1)}\|} \right)$. These values are used to produce DDIM samples.

To generate interpolations on a grid, we sample four latent variables and separate them in to two pairs; then we use slerp with the pairs under the same $\alpha$, and use slerp over the interpolated samples across the pairs (under an independently chosen interpolation coefficient). We show more grid interpolation results in Figure 11 (CelebA), Figure 12 (Bedroom), and Figure 13 (Church).

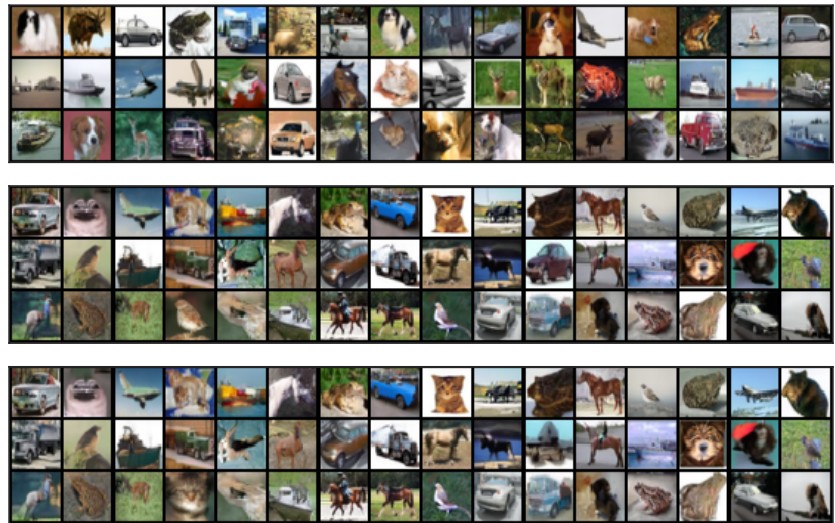

Figure 7: CIFAR10 samples from 1000 step DDPM, 1000 step DDIM and 100 step DDIM.

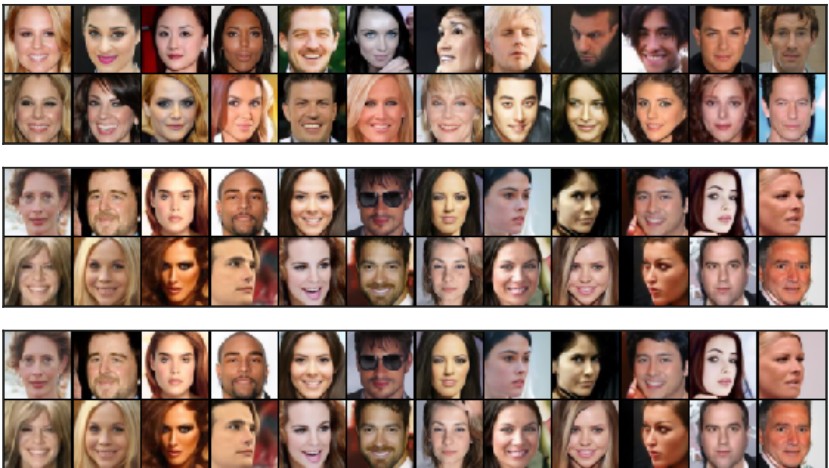

Figure 8: CelebA samples from 1000 step DDPM, 1000 step DDIM and 100 step DDIM.



Figure 9: CelebA samples from DDIM with the same random $x_T$ and different number of steps.

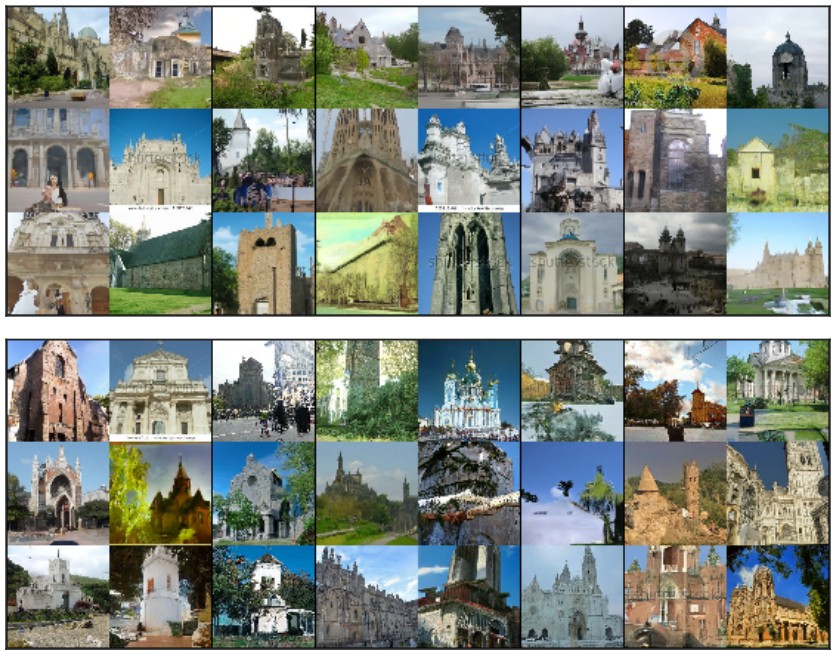

Figure 10: Church samples from 100 step DDPM and 100 step DDIM.

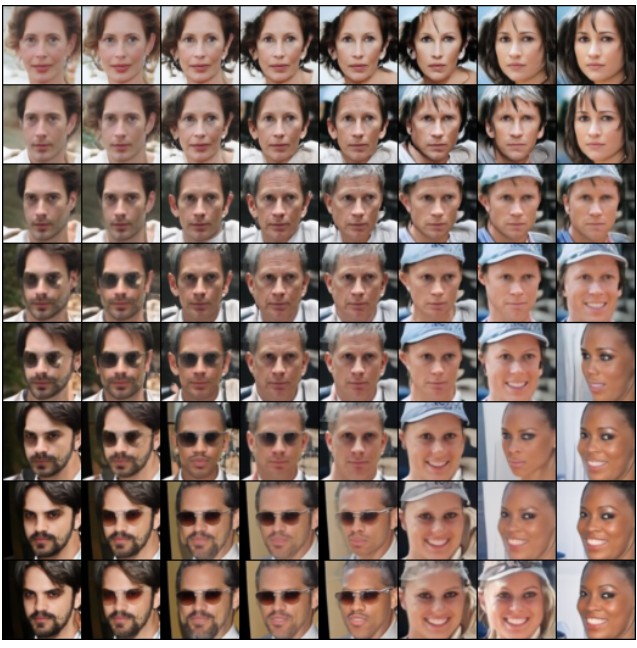

Figure 11: More interpolations from the CelebA DDIM with $\dim(\tau) = 50$.

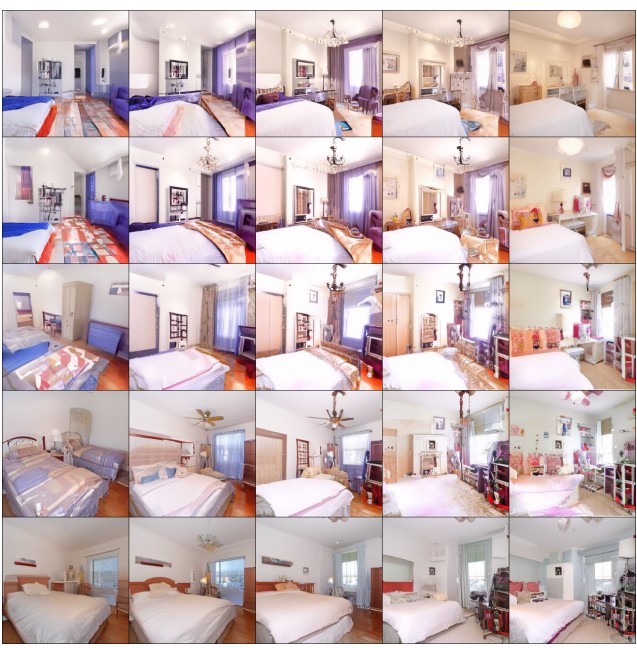

Figure 12: More interpolations from the Bedroom DDIM with $\dim(\tau) = 50$.

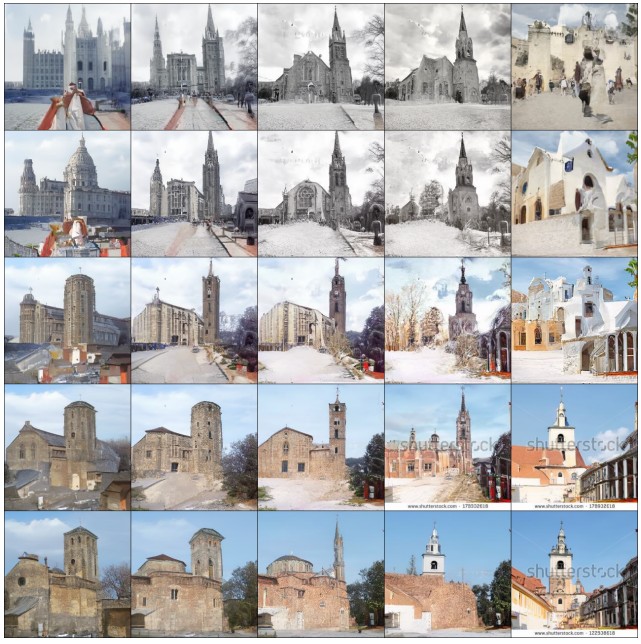

Figure 13: More interpolations from the Church DDIM with $\dim(\tau) = 50$.

