# OpenReview forum: "Denoising Diffusion Implicit Models"
_ICLR.cc/2021/Conference — ICLR 2021 Poster_

### Official Review · AnonReviewer1 · 2020-10-27
**An interesting approach but difficult to disentangle the explicit contributions.**

**Rating:** 6
**Confidence:** 3

**Review:**

#### DESCRIPTION
This paper consider tweaks to denoising diffusion models, exploring non-Markovian inference models, as well as shorter and possibly deterministic generative trajectories.

#### DISCUSSION

I'm having difficulty placing this work in the correct context, and disentangling exactly what the contributions are. From my understanding of denoising diffusion models (as presented in Sohl-Dickstein et al (2015) and Ho et al (2020)), the following hold:

(i) At test time, the variance of each step of the generative process can be set to a different value than at training time, effectively evaluating a different model than the one that was trained. This is exploited by Ho et al (2020) to remove the weights in the variational lower bound at training time (they effectively set sigma such that the weights are 1 at training time).

(ii) At test time, it is possible to select a subset of the timesteps used at training time, and generate a corresponding schedule for those selected timesteps, possibly leading to more efficient sample generation at the expense of sample quality.

(iii) Any inference process which satisfies the Markov property (as in Sohl-Dickstein et al or Ho et al) admits the factorization given in eq. (6).

Now, I would have thought the first port of call would be to compare the proposed method in this paper to (i) and (ii). Moreover, while I *think* the choice of q(xt-1 | xt, x0) in eq. (7) indeed means that q(xt | xt-1, x0) =\= q(xt | xt-1) (i.e. the process which starts at data and ends at noise is indeed non-Markovian), the lack of the comparisons with (i) and (ii) makes it unclear whether this step actually needs to be introduced, and the fact that the parameter sigma appears in both the mean and variance of q(xt | xt-1, x0) ties the mean and variance in a somewhat opaque way.

I'm also a little confused by the choice of language in the paper. Traditionally, a diffusion process describes a continuous-time stochastic process which satisfies the Markov property. 'Denoising diffusion models' stretch the term diffusion to describe a finite discretization of this diffusion process, but explicitly retain the Markov property, and have a clear interpretation as a generative models which iteratively denoises a white noise sample. If you further make the inference process non-Markovian, so that you have a non-continuous, non-Markovian process, does it make sense to persist in using the word diffusion to describe the model?

This may be a nitpick, but I'm also not sure about the sense in which this model can be described as 'implicit'. The term implicit is typically used to describe generative models which specify the data-generating procedure mechanistically as a series of prescribed operations (possibly stochastic) applied to some given inputs, so that the resulting output distribution is defined 'implicitly' by these generative steps, and *crucially* marginalizing over any associated latent variables is intractable. The 'implicit' model described in this paper is formulated as the deterministic limit of a standard latent variable generative model fit using amortized stochastic variational inference. Recent work has examined normalizing flows as the deterministic limit of a Markovian generative model (e.g. "Stochastic Normalizing Flows", Wu et al (2020), "SurVAE Flows", Nielsen et al 2020) and indeed, the sentence "The resulting model becomes an implicit probabilistic model, where samples are generated from latent variables with a fixed procedure." from this paper could equally describe a standard normalizing flow, which is certainly not an implicit model.

As far as the experiments are concerned:

- It seems the takeaway message from section 5.1 is that the proposed model better adapts to evaluation with shorter trajectories when eta is small, and that the original DDPM doesn't fare well with shorter trajectories. The sample quality gets worse the shorter the trajectory (which is expected), but the best results are still given by evaluating the full DDPM model with 1000 steps.

- In section 5.2, I suppose it's expected that samples generated using different length trajectories from the same starting point are similar given that the generative process likely corresponds to some ODE discretization (as you've mentioned in the conclusion), but it's good to have it tested.

- In section 5.3, I don't really understand the purpose of interpolations for demonstrating the capabilities of generative models. What exactly do they demonstrate?

#### EXTRA NOTES

Why are 'joint' and 'marginals' repeatedly placed in quotation marks?

"The magnitude of σ controls the how stochastic the forward process is.", "...the stochasticity of the process", "...compared to its less stochastic counterparts" What does it mean for something to be more or less 'stochastic' than something else? How can 'stochasticity' be used as a quantitative measure?

"Fenchel Inception Distance" -> "Frechet Inception Distance"

#### CONCLUSION
Overall, I think fundamental idea of improving the long sampling trajectories of DDPMs is certainly interesting. However, I'm finding it difficult to understand the extent to which the proposed non-Markovian inference process is really necessary i.e. I'd like to see baseline comparison with points (i) and (ii) above. Right now, I'm not able to disentangle which components of the proposed method are actually necessary.

---

> ### Author Response · Authors · 2020-11-20
> **We make diffusion type models much faster and easier to apply in real-world scenarios, whereas existing methods cannot.**
>
> We start by clarifying that our main contribution is not to outperform DDPMs under 1000 steps; as we argue in the introduction, these many steps cost too much compute, so as it stands, DDPMs are unlikely to be used in any practical, latency-sensitive scenarios.
>
> Our contributions are twofold:
> - On a practical level, we propose methods that are computationally more efficient by 10x to 50x, while retaining the sample quality of the original case (see Figure 5). We show in the paper and in the following response that a compute-limited variant of DDPM does not perform nearly as well, even with the modifications suggested by the reviewer.
> - We are the first to introduce non-Markovian forward processes that generalize the Markovian ones from the literature, which enable a large family of generative models, including categorical ones for discrete data (Appendix A in revision).
>
> **Q: Compare with two possible modifications to DDPM: (i) set variance of each step of the generative process differently; (ii) select a subset of timesteps and generate corresponding schedule for selected timesteps.**
>
> **A**: **We explicitly considered (ii) in the last 2 rows of Table 1 (original version and revised version)**, where we used fewer timesteps for DDPM (from T=0 to 100) to achieve comparable runtimes / accelerate sampling; the fact that sampling can be made faster is not explicitly mentioned in (Ho et al. 2020), but we count it as a DDPM baseline nonetheless. We find that our approach generates better samples than DDPM in the accelerated cases.
>
> For (i), we did an additional experiment where we set the variance in the DDPM generator to zero, where we keep the mean function the same as in (Ho et al. 2020). **This results in images that are entirely gray, corresponding to samples that contain all zero values.** This is not unexpected, because it represents a case where the generation process is a poor approximation to the reverse of the Markovian inference process.
>
> **Q: What components of DDIM are necessary for good sampling quality in a shorter time?**
>
> **A**: Comparisons with (ii) show that having a smaller variance in the sampling process is necessary. Comparisons with (i) show that simply having a smaller variance in DDPM is not enough; we must also revisit the mean function accordingly, which only comes from our generalized non-Markovian forward process analysis.
>
>
> **Q: Why is the “non-Markovian” point of view useful?**
>
> **A**: As we have shown with the comparisons of (ii) and (i), without our contributions (non-Markovian) inference process, the corresponding generative process fails to recover high-quality samples in a shorter time. In Appendix A of the revision, we also show how the “non-Markovian” view can be generalized to discrete inference distribution as well.
>
> **Q: Why call the method “diffusion”?**
>
> **A**: In the abstract of (Sohl-Dickstein et al., 2015), they mentioned “we then learn a reverse diffusion process”, but the forward process in their case is also just a discretization of a diffusion process. We keep the term “diffusion” to highlight the connections to existing works that work under discretized time.
>
> **Q: “This may be a nitpick, but I'm also not sure about the sense in which this model can be described as 'implicit'.” (“The term implicit … and crucially marginalizing over any associated latent variables is intractable.”)**
>
> **A**: We follow the definition from a line of work on “implicit generative models”.
>
> Regarding modeling assumptions, we copy this quote **verbatim** from [Mohamed and Lakshminarayanan, 2016]:
> "Implicit generative models use a latent variable $z$ and transform it using a deterministic function $G_\theta$ that maps from $R^m \to R^d$ using parameters $\theta$."
>
> For our deterministic sampling process, our latent variable is $x_T$, our process is a deterministic function that maps from $R^d \to R^d$, so it is an implicit generative model in this sense. It differs from DDPM in the sense that the DDPM mapping from $x_T$ to $x_0$ is not deterministic.
>
> Regarding tractability, we copy this quote (also **verbatim**) from [Diggle and Gratton, 1984]:
> "In principle, a unique log-likelihood function underlies any such implicit statistical model (but not vice versa) and **if this can be written down explicitly**, inference can proceed along conventional lines."
>
> The fact that likelihood can be tractable does not make a model less implicit; by this argument, normalizing flows are also implicit models (just with tractable log-likelihood functions). In fact, Flow-GAN [Grover et al., 2018] trains a flow with adversarial training, which can be advantageous for some tasks other than density estimation.
>
> [Mohamed and Lakshminarayanan, 2016], Learning in Implicit Generative Models
> [Diggle and Gratton, 1984], Monte Carlo Methods of Inference for Implicit Statistical Models
> [Grover et al., 2018] Flow-GAN: Combining Maximum Likelihood and Adversarial Learning in Generative Models

---

> > ### Author Response · Authors · 2020-11-20
> > **cont'd**
> >
> > **Q: “the best results are still given by evaluating the full DDPM model with 1000 steps.”**
> >
> > **A**: First, the goal of our paper is to drastically reduce sampling time while retaining good sample quality, not to optimizeFID scores with as much compute as possible. Second, the DDPM model uses different variances for different datasets to achieve the best score; we mentioned this in the experiment section that CIFAR10 used a **larger variance than LSUN**, and even included the [exact line of code from Ho et al.’s implementation](https://github.com/hojonathanho/diffusion/blob/master/scripts/run_cifar.py#L136) in our first version. From our experiments, if the LSUN variance is used for CIFAR10 then worse FID scores will be observed. In contrast, we do not tune this extra hyperparameter and still achieve good performance.
> >
> > **Q: Why do the interpolation experiments at all?**
> >
> > **A**: It reveals an interesting property of DDIM that GANs also have, so there is the potential to use them for applications such as image manipulation. DDPMs, on the other hand, do not have this property.
> >
> >
> > **Q: Why are 'joint' and 'marginals' repeatedly placed in quotation marks?**
> >
> > **A**: In the update we have removed the quotation marks and commented on how this may be a slight abuse of notation.
> >
> >
> > **Q: How can 'stochasticity' be used as a quantitative measure?**
> >
> > **A**: We use the term ‘stochastic’ to refer to larger $\sigma_t$ values, since we can choose a different $\sigma$ for each $t$, we simplify the case by using a scalar coefficient $\eta$ that measures how much noise we inject at each step. If $\eta = 0$, then the process is deterministic (i.e., least “stochastic”), if $\eta = 1$, then we inject more noise at each step. Our empirical results (Figure 3) support these claims.
> >
> > Thank you for the review.

---

> > > ### Comment · AnonReviewer1 · 2020-11-22
> > > **Response**
> > >
> > > Re use of language ('diffusion', 'implicit', 'stochastic' etc.): I think it's important to choose language carefully, because clear communication of an idea can often be as important as the idea itself.
> > >
> > > Despite your method's clear influences, since a diffusion typically refers to a continuous-time Markov process, my question is whether the term diffusion is really the best way to describe a discrete-time non-Markovian process.
> > >
> > > "Regarding modeling assumptions, we copy this quote verbatim from [Mohamed and Lakshminarayanan, 2016]"
> > > OK, but consider the following from the remainder of the paragraph from which that quote is taken:
> > > "When the function G is well-defined, such as when the function is invertible, or has dimensions m = d with easily characterised roots, we recover the familiar rule for transformations of probability distributions. We are interested in developing more general and flexible implicit generative models where the function G is a non- linear function with d > m, specified by deep networks. The integral (2) is intractable in this case: we will be unable to determine the set {Gθ (z) ≤ x}, the integral will often be unknown even when the integration regions are known and, the derivative is high-dimensional and difficult to compute. Intractability is also a challenge for prescribed models, but the lack of a likelihood term significantly reduces the tools available for learning. In implicit models, this difficulty motivates the need for methods that side-step the intractability of the likelihood (2), or are likelihood-free."
> > > Intractability of the likelihood is one of the defining factors of an implicit model, evidenced by the fact that the terms implicit and likelihood-free are often used interchangeably, and the fact that the above paper exists to deal with learning in implicit generative models *because* the likelihood is intractable. Your model has a tractable lower bound on the likelihood which you use for training, and only becomes deterministic at test time in the limit of a scalar hyperparameter. I also do not understand how a normalizing flow could be described as an implicit model.
> > >
> > > "how stochastic", "vary the... stochasticity", "less stochastic"
> > > I repeat: 'stochasticity' is *not* a quantitative measure. What you're trying to describe is the entropy of the process, which *is* a quantitative measure. Saying one stochastic process is more stochastic than another doesn't have meaning.
> > >
> > > Re contributions: I think the fundamental idea of the paper, namely improving on the long sampling time of DDPMs using some model tweaking, is definitely worthwhile. I'm inclined to believe that the proposed tweaks on the DDPM model are justified, and the experimental results certainly suggest an improvement in wall-clock time, even at the expense of some quality.
> > >
> > > Ultimately, it seems as if the paper has been pitched as a more efficient DDPM-style approach, in terms of language, theory, and empirical evaluation. If the goal is to "make diffusion type models much faster and easier to apply in real-world scenarios", then the gold standard comparison is a GAN (qualitative differences in model class notwithstanding), but there's no comparison there. Overall, I feel comfortable raising my score to a 6.

---

### Official Review · AnonReviewer3 · 2020-10-28
**Very good, clean, paper**

**Rating:** 8
**Confidence:** 4

**Review:**

Summary : This paper develops a variant (DDIM) of an existing method (DDPM) with the goal of accelerating it greatly while still maintaining performance. The authors are working in the context of a denoising process that runs in the reverse direction to a sequence of steps that each add a small amount of Gaussian noise to the original data. The proposal is to introduce an auxiliary function that breaks the Markov assumption by leaking some information in a controlled way about the training points x0, and then use this auxiliary function as scaffolding to train the actual Markov chain of denoising functions.

This paper builds on other works in the recent literature and proposes something useful and novel. They propose something relatively down-to-earth and then they methodically analyze the consequences and derive all the mathematical formulas that follow. I feel that the dose of mathematical content is just appropriate for what they set out to do. Less would be too vague, more would be excessive.

The direction that they are proposing is relevant, contrary to certain other papers that involve a lot of correct equations but don't take us anywhere interesting. In my mind, this is a very good clean paper.

I appreciated the authors taking the time to mention the connection with ODEs, and I thought they would mention Neural ODEs in section 7 as a nod to the recent surge of interest in ODEs in the context of Deep Learning sparked by that paper.

The only bad things that I could possibly say about this paper would involve comparing it to certain other wildly creative papers, and to say that it's not as innovative or throught-provoking as those papers. And that's not a fair thing to say.

I would like to ask the authors if, with their framework, there is a need to train a completely different epsilon_t for every t=1..T ? I understand these models to be U-Nets, as mentioned in the appendix. I presume that the thing that makes this reasonable is the fact that S < T, and only S different models need to be trained?

---

> ### Author Response · Authors · 2020-11-20
> **There is no need to train a different epsilon_t**
>
> Thank you for the review. In Appendix A of the revision, we show how our idea can be generalized to discrete / multinomial inference distribution as well, which resulting in an objective that does not minimize MSE but rather minimizes classification error.
>
> **Q: Is there a need to train a different epsilon_t?**
>
> **A**: In our experiments, we trained using the same maximum T (1000) and then chose different sampling processes $S$ accordingly.
>
> If it is known a-priori that we will at most sample in S steps as well as which $\alpha_t$ we will choose, then having more $\epsilon_t$ than S seems unnecessary. However, if we train with more steps, then we have greater flexibility in trading-off computational costs for better sample quality / FID scores. Moreover, some schedules for $\alpha_t$ might work better than others, so training with more steps than what is needed for sampling seems adequate (the model size does not increase much even with more steps, due to the positional embeddings that condition the step).

---

### Official Review · AnonReviewer2 · 2020-10-28
**Keen insights, good results**

**Rating:** 7
**Confidence:** 4

**Review:**

Summary:

This paper proposes a change to the recently popular diffusion models, motivated by increasing the speed of sampling. This is accomplished by changing the “forward” process which adds noise to the data. In the original diffusion models, this forward process is a Markov process whose marginals and conditionals can be computed efficiently in closed form. This paper proposes to replace this Markov forward process with a non-markovian process that is designed to have the same marginals. The generative model, in this case, changes such that to predict the next step in the process, the model must first predict the “clean” sample at the end of the chain which is then used to give an estimate for the next step in the chain.

Intriguingly, the objective for training this new generative model is identical to training a standard diffusion model. Thus, the models differ only at sampling time. Under the new interpretation, we can sample from a family of models after training. This family can be tuned to increase the speed of sampling, at the cost of some sample quality. Inside this family, there also exists an implicit generative model which can be sampled from in a more deterministic fashion than the other members -- hence the name of the paper.

The authors present a number of image generation experiments and study the impact of the sampling parameters on sample quality. They find that in settings where the efficiency of sampling is greater than the original diffusion model, the proposed approach achieves higher sample quality -- although never as high as the original diffusion model.

Because the model can be sampled in a deterministic fashion, the authors note that fixing the latent variables during sampling results in consistent samples. This allows a much more controlled generation compared to the original diffusion model.


Strong areas:

I quite enjoyed this paper. These diffusion models have gained considerable attention and this work addresses what is likely their largest issue -- the slow speed of sampling. Most interesting to me is these new ideas can be applied to the original diffusion models without retraining.

I find that this work provides more insight into how these diffusion models work and which choices in the original presentation are important for their performance and which (like the inference process) can be changed. These are important insights and should be impactful for further research on this class of models.


Weaknesses:

While this is a strong paper, there are a few issues in my opinion. I found some of the experimental details difficult to understand. Were all the results presented generated from the same DDIM? Or were the models trained separately for each sampling parameter set? This should be made more clear.

In section 3.2 you claim some equivalence between the original DDPM objective and the variational bound in your model. This looks like re-weighting the various terms in the objective. In the DDPM work, they discuss the impact of this re-weighting. I would have liked to see a similar discussion here. They found it improved sample quality but decreased likelihood. I would have loved a comparison of the various training procedures but I did not find one. Along the same line, you present no likelihood results. You should be able to generate a lower-bound on likelihood using your model (except for the eta=0 model I believe). I am curious how the various sampling schemes impact likelihood. It is not necessary for the proposed approach to be appealing but would definitely complete the picture.


Nit-picky issues:

I think the abstract could be a bit more descriptive. You should add some information saying what differs between your approach and standard diffusion models. In reading the abstract, it was not clear to me that using a non-markovian forward process would make sampling faster. This became clear as I read the paper. You could add a little more information to make that clear in the abstract.


My recommendation:

I think this paper was clearly written and proposed a strong contribution to the field of generative modeling. I think the insights presented here give us more understanding of diffusion models and while also improving their sampling speed. I will recommend accepting this paper.

Some questions:

Do you think these same insights could be applied to discrete diffusion models proposed in the original work on non-equilibrium thermodynamics?

---

> ### Author Response · Authors · 2020-11-20
> **DDIM uses the same model for all experiments, and can generalize to discrete cases**
>
> Thank you for the review.
>
> **Q: Were all the results presented generated from the same DDIM?**
>
> **A**: Yes, we use the same model for all the experiments within each dataset. In particular, all the models are trained with $L_{simple}$ objective in (Ho et al., 2020); apart from CelebA 64x64 (where we do not have pre-trained checkpoints), all other experiments use the checkpoints from (Ho et al., 2020). We emphasize this point in the revision.
>
> **Q: How would different training procedures affect performance?**
>
> **A**: As we mentioned in Theorem 1, an advantage of the non-Markovian perspective is that it shows the equivalence of using any weights $\gamma_t$ at different time $t$ (at least in principle, if the parameters are not shared). In practice, using the same weight across different objectives may be advantageous because the Adam optimizer keeps track of the gradient at each step to better condition the problem, and similar weights could be more stable simply from an optimization perspective.
>
> **Q: Do you think these same insights could be applied to discrete diffusion models proposed in the original work on non-equilibrium thermodynamics?**
>
> **A**: While our focus on the paper is not exactly for discrete models, we believe that discrete diffusion models could also benefit from such a view. In Appendix A of our revision, we discuss in detail how we can derive the variational objective for categorical data with **multinomial forward processes**, under various levels of stochasticity.
>
> Analogous to how the objective for the Gaussian case minimizes MSE, **the objective for the multinomial case minimizes multi-class classification error** (with the target being the one-hot vector observation). Similar to the Gaussian case, we can use an objective with the same weights for each step, and choose different sampling processes.
>
> **Q**: What would the lower bounds to the likelihood with non-Markovian procedures become?
>
> **A**: In the derivation of Theorem 1, we showed that the variational lower bounds to the log-likelihood are inversely correlated with the $\sigma$ parameters, so the bounds can only become worse as we decrease $\sigma$. Intuitively, this is because unlike DDPM, the forward (inference) process is no longer a good approximation to the posterior of the backward (generation) process. Nevertheless, we show in the revised Section 5.4 that we can find latent $x_T$ that reconstruct $x_0$ in the test set relatively well.
>
> **Q: Abstract can be more descriptive.**
>
> **A**: We have updated the abstract to reflect that the non-Markovian aspect is what makes it possible to derive the deterministic sampling method.

---

> > ### Comment · AnonReviewer2 · 2020-11-22
> > **Thanks for the response**
> >
> > Authors,
> >
> > Thank you for responding to my comments and clarifying my understanding. I am glad that you have made the changes you did to the paper. I believe these changes have slightly improved the work. I initially reviewed this paper at a 7 and I do not feel these new changes warrant me to further raise this score, so I will stay at a 7. This is a good paper and I hope it is accepted.

---

### Decision · Program_Chairs · 2021-01-07
**Final Decision**

**Decision:**

Accept (Poster)

**Comment:**

This work provides additional insights into a class of generative models that is rapidly gaining traction, and extends it by potentially providing a faster sampling mechanism, as well as a way to meaningfully interpolate between samples (an ability which adversarial models, currently the most popular class of generative models, also have). The revised manuscript includes an extension to discrete data, which could potentially amplify the impact of this work. The authors have also run additional experiments in response to the reviewers' comments.

Reviewer 1 raised several concerns about the choice of language (i.e. referring to the proposed model as a diffusion model, and the precise meaning of 'implicit' in the context of generative models). This is a fair point, as the authors introduce changes that affect the Markovian nature of the "diffusion" process, and a diffusion process is supposed to be Markovian by definition.

However, I think there is something to be said for the authors' argument of using the word 'diffusion' to clearly link this work to the prior work on which it is based. Given that technically speaking, the original DDPM work already 'abuses' the term to refer to a discrete-time process, it is difficult to argue compellingly that 'diffusion' should not feature in the name of the proposed model. Referring to 'non-Markovian diffusion processes' however seems more problematic, as this is a direct contradiction. If the authors wish to use this phrase, adding a few sentences to the introduction that justify this use would be helpful, and personally I feel this would be sufficient to address the issue (I noted that Section 4.1 already acknowledges that the forward process is no longer a diffusion). Plenty of work in our field abuses notation and this is justified simply with the phrase "with (slight) abuse of notation..."; I don't think this would be any different.

Reviewer 1 is technically correct that 'stochastic' is an absolute adjective, i.e. something can only be stochastic or deterministic, there is nothing in between, and there are no degrees/levels of stochasticity or determinism. In practice however, it is quite often used in a comparative sense, and I believe I have in fact been guilty of this myself! I do not feel that it causes any ambiguity in this case. Indeed, the phrase 'degree of stochasticity' seems to be in relatively common use in literature. While there may be more correct terms to use, I subscribe to the descriptivist view on language, and I do not think the comparative use of 'stochastic' is a major issue here. The alternatives I can think of seem potentially more cumbersome (e.g. I wager that 'more/less entropic' would be more poorly understood than 'more/less stochastic'). Still, I recommend that the authors consider potential alternatives in the future, to avoid any confusion.

Overall, I think the reviewers' major concerns have been addressed in the revised manuscript. Given that all reviewers consider the idea worthwhile, I will join them in recommending acceptance.